Brief Communication

# A realistic phantom dataset for benchmarking cryo-ET data annotation

Ariana Peck[1,3], Yue Yu [1,3], Jonathan Schwartz[1,3], Anchi Cheng [1], Utz Heinrich Ermel [1], Joshua Hutchings[1], Saugat Kandel [1], Dari Kimanius [1], Elizabeth A. Montabana[1], Daniel Serwas[1], Hannah Siems[1], Feng Wang[2], Zhuowen Zhao [1], Shawn Zheng[1], Matthias Haury [1], David A. Agard[1,2], Clinton S. Potter[1], Bridget Carragher [1], Kyle Harrington [1]✉ & Mohammadreza Paraan [1]✉

Cryo-electron tomography (cryo-ET) is a powerful technique for imaging molecular complexes in their native cellular environments. However, identifying the vast majority of molecular species in cellular tomograms remains prohibitively difficult. Machine learning (ML) methods provide an opportunity to automate the annotation process, but algorithm development has been hindered by the lack of large, standardized datasets. Here we present an experimental phantom dataset with comprehensive ground-truth annotations for six molecular species to spur new algorithm development and benchmark existing tools. This annotated dataset is available on the CryoET Data Portal with infrastructure to streamline access for methods developers across fields.

Much of cell biology is still uncharted territory. Cryo-electron tomography (cryo-ET), a method used in structural cell biology, is uniquely poised to expand our understanding of cellular function in health and disease[1]. In cryo-ET, samples are cryopreserved to maintain the structural integrity of cellular components and provide a detailed view of the cellular state at the moment of freezing[2]. A region of interest (ROI) is illuminated from different orientations with an electron beam, and the corresponding two-dimensional (2D) projection images are collected (Fig. 1a, top). These 2D images are then aligned and reconstructed to generate a tomogram, which is a three-dimensional (3D) map of the ROI (Fig. 1a, bottom). This imaging modality provides both the fine sampling required to obtain near-atomic-resolution (3–4 Å) structures of molecular complexes and the field of view (a few hundred nm to a few µm) needed to visualize these complexes' cellular context[3–6].

However, there are several challenges to high-resolution in situ structure determination. In cryo-ET, the range of projection views is restricted to ±60°, giving rise to a missing-wedge artifact[7] in the reconstructed volume that smears information along the imaging axis. Additionally, the radiation sensitivity of biological samples severely restricts the dose that can be applied, resulting in data with

very low signal-to-noise ratios (SNRs)[1]. Consequently, generating a high-resolution map of a molecular species often requires identifying, aligning and averaging tens of thousands of individual copies of the target molecule across hundreds of tomograms.

The process of identifying individual copies of molecules in tomograms is referred to as annotating, picking or labeling. Annotation remains a significant bottleneck owing to tomograms' low SNRs, the structural complexity of cellular samples, the diversity and heterogeneity of the molecules of interest and the large numbers of molecules required for high-resolution maps[8]. In most cases, annotation is the most time-intensive and laborious part of cryo-ET data processing because it often relies heavily on manual input. Given that comprehensive labeling is critical to both obtain high-resolution structures of molecular complexes and reveal large-scale ultrastructure, there is an urgent need for new methods to annotate cellular tomograms at scale.

ML algorithms are well-suited to overcome this bottleneck[9–13]. Compared with the conventional method of template matching, ML can deliver more efficient and versatile solutions. At the time of writing, >19,000 tomograms are publicly available on the recently launched CryoET Data Portal[14]. ML has been used to provide membrane

[1]Chan Zuckerberg Institute for Advanced Biological Imaging (CZ Imaging Institute), Redwood City, CA, USA. [2]Department of Biochemistry & Biophysics, University of California, San Francisco, CA, USA. [3]These authors contributed equally: Ariana Peck, Yue Yu, Jonathan Schwartz. ✉e-mail: kharrington@chanzuckerberg.com; reza.paraan@czii.org

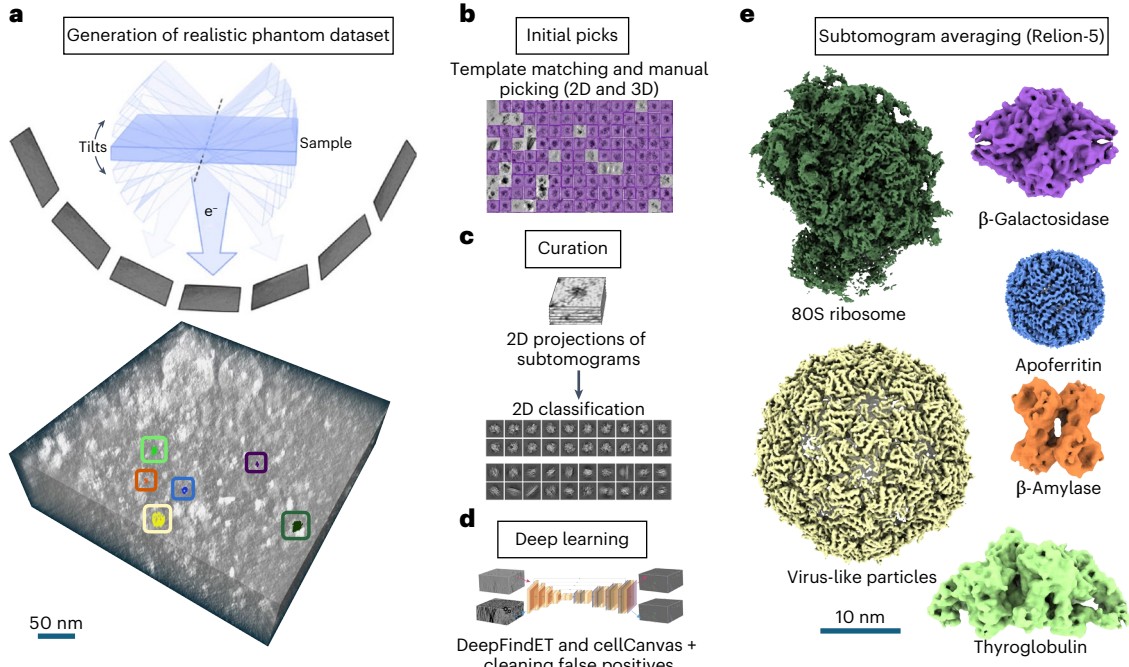

**Fig. 1 | Creating and annotating a phantom dataset. a**, Top, a schematic of the tilt-series data collection process for generating 3D volumes in cryo-ET, demonstrating angles from +45° to −45° for the phantom dataset. Bottom, a phantom tomogram, in which different colors represent different class labels, following the color code in **e. b**–**d**, Overview of the workflow for generating ground-truth labels. In this figure, 2D refers to the slab picking and the minislab approach, in which the tomograms and subtomograms, respectively, are projected to 2D along the *z* axis for processing; 3D refers to the pipelines that use 3D tomograms and subtomograms as input. **b**, Initial picks were generated using PyTom 3D template matching[43,44], 2D slab picking and manual picking. The copick format was used to standardize the metadata across multiple workflows. **c**, The initial picks were curated after extraction and projection to 2D images, called the minislab method. These were processed in CryoSPARC[45] in a single-particle-analysis pipeline with a pixel size of 5 Å per pixel, leveraging 2D classification, Topaz[12] and manual picking for curation. **d**, The curated picks and labeled synthetic tomograms were used as a training set for DeepFindET and CellCanvas, which resulted in a tenfold increase in the number of picks per species, which were subsequently cleaned of false positives. **e**, The final curated picks were refined using a Relion-5 pipeline to provide maps with resolutions of: ribosome (4.3 MDa), 3.6 Å, 24,338 particles; VLP (3.4 MDa), 4.1 Å, 3,022 particles; THG (660 kDa), 8.6 Å, 6,211 particles; β-galactosidase (540 kDa), 7.8 Å, 3,113 particles; apoferritin (450 kDa), 3.9 Å, 23,393 particles; β-amylase (268 kDa), 11.5 Å, 2,464 particles.

segmentations for all of these datasets[13,15–17], but labeling individual molecules, or particles, is much more difficult because of their diversity, lower contrast and cellular crowding. As a result, only one-fifth of the deposited tomograms have molecular annotations. Algorithms that scale are needed to fully leverage this wealth of unannotated data and keep pace with increasingly rapid data acquisition. In addition to scale, versatility is critical so that labeling strategies can be adapted across datasets. To date, ML particle picking methods have been effective for specific use cases[9–13] but do not generalize sufficiently well to meet the diverse needs of the cryoET community.

Benchmark datasets prepared from standard samples have proven essential to critically evaluate state-of-the-art algorithms and spur innovative development[18–21]. In the biomedical research community, physical models of biological tissue used to validate imaging methods are referred to as phantoms[22]. Adopting this terminology, we designed a phantom sample to enable a rigorous assessment of annotation algorithms. To capture artifacts characteristic of cryo-ET data, including the noise behavior, residual beam-induced motion and the missing wedge, we designed an experimental, rather than synthetic, sample.

We have released this experimental annotated phantom dataset of nearly 500 tomograms to spur the development and rigorous benchmarking of novel ML algorithms for cryo-ET particle picking. To encourage generalizability, we have selected six target particles with diverse shapes, collectively spanning an order of magnitude in molecular weight (Fig. 1 and Table 1). Five of these particles—virus-like particles[23] (VLPs), thyroglobulin[24] (THG), apoferritin[25], β-galactosidase[26] and β-amylase[27]—were mixed with cellular lysate, which contains the sixth particle—80S ribosomes[28]—in abundance (Extended Data Fig. 1). The lysate contains numerous non-target particles and structural elements, such as membranes, which often confound annotation

**Table 1 | Molecular species (from different sources) present in the phantom dataset**

| Sample | Molecular weight | Symmetry/copies | EMDB ID |
|---|---|---|---|
| Ribosome[28] (80S; human) | 4.3 MDa | Monomer 1× | EMD-3883 |
| VLP[23] (PP7 bacteriophage) | 3.4 MDa | Icosahedral 60× | EMD-41917 |
| Thyroglobulin[24] (bovine) | 660 kDa | Homodimer 4× | EMD-24181 |
| Beta-galactosidase[26] (*Escherichia coli*) | 540 kDa | D2 4× | EMD-0153 |
| Apoferritin[25] (equine) | 450 kDa | Octahedral 24× | EMD-41923 |
| Beta-amylase[27] (sweet potato) | 268 kDa | D2x 4× | EMD-30405 |
| Albumin[42] (HSA) | 66 kDa | Monomer 1× | EMD-43090 |

The chosen annotation targets spanned a large range of molecular weights and distinct symmetries. The Electron Microscopy Data Bank (EMDB) entries were used for initial picking with PyTom[43,44] template matching (excluding HSA, which was not an annotation target).

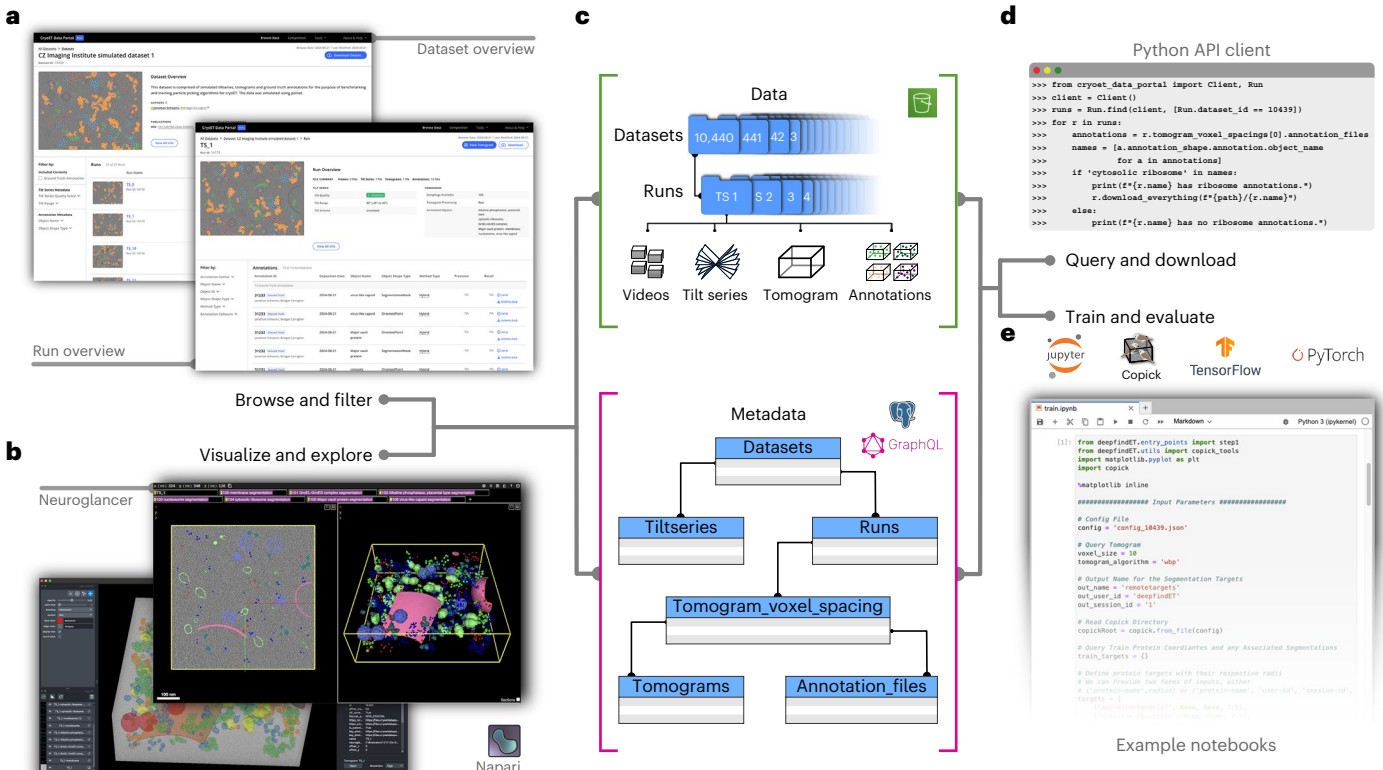

**Fig. 2 | The CryoET Data Portal and related infrastructure. a**, The CryoET Data Portal[14] web interface allows users to explore the experimental and simulated phantom datasets (https://cryoetdataportal.czscience.com/depositions/10310; dataset IDs 10440, 10441, 10445, 10446), including tomograms, annotations and associated metadata (CC0 license). **b**, Tomograms and annotations can be visualized directly inside the web browser using Neuroglancer, or in the desktop viewer napari[37]. **c**, All related image data and metadata are provided from a public AWS S3 bucket, and associated metadata can be queried using a publicly exposed GraphQL API. **d**, Both resources can be accessed programmatically using the CryoET Data Portal python API client, which also provides convenient methods for data download. **e**, All CryoET Data Portal resources (including other datasets) can be accessed using copick and the provided example notebooks, which feature reference implementations in TensorFlow and PyTorch.

algorithms (Extended Data Fig. 2). Although less crowded than a cellular sample, this dataset provides realistic artifacts inherent to experimental tomograms and enables high-confidence labeling. We generated comprehensive, meticulously validated ground-truth labels for the target species using elaborate workflows that yielded new annotation tools but underscored the need for streamlined solutions. This phantom dataset has already served as a testbed for a new annotation algorithm[29] developed by cryo-ET experts and as the foundation of a Kaggle competition that engaged ML experts from beyond the field.

To emulate the cellular environment, a key component of this sample was cellular lysate selectively enriched for lysosomes (Extended Data Fig. 1 and Methods). These organelles established a sample thickness of ~200 nm, aligning with the target thickness of most cellular lamellae[30,31] (Extended Data Fig. 3). This thickness allowed multiple layers of molecules to be stacked along the imaging axis, making annotation more challenging than for single-particle cryo-EM datasets, which usually feature a monolayer of particles. The lysosome-capture protocol was intentionally mild to retain abundant molecular species, including ribosomes. This enriched lysate was supplemented with high concentrations of five target species (Fig. 1e and Table 1), in addition to human serum albumin (HSA) to increase molecular crowding. Visual inspection suggested that HSA and other small, non-target molecules consistently populated the air–water interface (AWI) and displaced target particles from this boundary (Extended Data Fig. 4). As a result, the species of interest do not suffer from AWI-induced preferred orientations[32] and are fully represented without angular sampling bias (Extended Data Fig. 5). In summary, the lysosome-enriched lysate was

crucial to mimic cellular cryo-ET data and bypass common limitations of plunge-frozen samples.

In addition to capturing the imaging artifacts in cryo-ET, the phantom dataset also contains significant heterogeneity that challenges annotation algorithms. The conformational landscape of ribosomes[33] was retained because this species was captured directly from cells. Another source of heterogeneity is the shapes of the six target molecules. Although distinct in 3D, some targets appear similar along certain 2D projections. For β-amylase, β-galactosidase and THG, in particular, projected views along the small dimension of the larger particle resemble some projections along the large dimension of the smaller particle (Extended Data Fig. 5). Some of the non-target particles derived from the lysate also appear similar to certain target species in projection and thus serve as natural decoys (Extended Data Fig. 2). Together, these factors raise the bar for picking algorithms to discriminate among particle classes.

The size and quality of this dataset are also important features. After visual inspection and curation on the basis of quality metrics[34], 492 of the 1,070 collected tomograms were retained (Extended Data Fig. 6). The remaining tomograms were then rigorously and comprehensively annotated by combining the new tools we developed for particle picking, curation and visualization with existing software (Supplementary Table 1, Extended Data Figs. 7 and 8 and Methods). The resulting workflows yielded ~60,000 high-quality labels across the six target species, which we refer to as the ground truth. These ground-truth labels were validated on the basis of the resolution of the 3D reconstructions (Fig. 1e and Extended Data Fig. 9) and the quality of the 2D classes (Extended

Data Fig. 5) of the annotated particles. However, these ground-truth annotations are incomplete given the experimental nature of the data. Despite rigorous curation, the lack of an absolute measure of particle quality at the individual copy level means that we cannot entirely rule out false positives, especially for smaller targets. And we have almost certainly missed labeling some true positives, which we anticipate future annotation algorithms will recover. Because new annotation sets can be easily uploaded to the CryoET Data Portal[14], we anticipate that this dataset's labels will be an evolving benchmark. However, the quality and quantity of the original ground-truth labels already provide a highly accurate training set and sufficiently large test set to rigorously benchmark annotation algorithms.

The phantom dataset is available on the CZ CryoET Data Portal[14] (CZCDP) under deposition ID CZCDP-10310. In this deposition, the phantom dataset is split into a training set (7 tomograms), a public test set (121 tomograms) and a private test set (364 tomograms). This deposition also includes a synthetic dataset (27 tomograms), generated by PolNet[35], of the target particles and simulated cellular components. This organization is inherited from an ML challenge (https://www.kaggle.com/competitions/czii-cryo-et-object-identification/) based on the phantom dataset so that consistent training and test data can be used to benchmark new and existing algorithms.

Regardless of domain expertise, researchers can familiarize themselves with the phantom data, as well as other cryo-ET datasets, by browsing the Portal's web interface and visualizing the ground-truth annotations in a web browser using a Neuroglancer[36]-based viewer (Fig. 2). Outside the browser, datasets can be queried using a Python-based API client and visualized using a napari[37] plugin. All datasets are accessible with a consistent layout and metadata schema in public cloud storage, and directly link raw data, tilt series, tomograms and annotations (Fig. 2). Processed image data are stored in the cloud-ready next-generation OME-Zarr[38] format, allowing researchers to train models on datasets exceeding available local storage by streaming tomograms whole or in part. PyTorch[39] and TensorFlow[40] infrastructure relying on these features is available through the direct integration of CZCDP datasets with copick[41] (Supplementary Table 1). Example notebooks that apply published algorithms to this dataset are provided so that developers can focus on model development rather than data handling (Supplementary Table 2).

The effectiveness of our phantom dataset for benchmarking was recently tested during a competition hosted on Kaggle (https://cryoet-dataportal.czscience.com/competition). Contestants were tasked with developing multi-class annotation algorithms after being provided with only seven training tomograms to mimic the volume of data that is practical for researchers to annotate by hand. Their submitted annotations were then scored against the remaining withheld tomograms. The top-scoring solutions, which are deposited on the CryoET Data Portal, outperformed our own best-effort model, DeepFindET (Methods), demonstrating the value of the phantom as a resource for developing new annotation algorithms and rigorously assessing performance. This benchmark dataset will be used to track progress and highlight limitations that still need to be overcome to achieve robust annotation of cellular tomograms at scale.

## Online content

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

## Methods

### Sample preparation

The phantom dataset was generated from a combination of cell lysates, commercially available purified protein and one purified protein obtained from collaborators (Table 1). The cell lysates were derived from a HEK293T cell line, generated at the Chan Zuckerberg BioHub by M. Leonneti's group, with a knock-in GFP-Tag on the carboxy terminus of the lysosomal house-keeping protein LAMP1 (ref. 46). The homogenization protocol[47] was developed at M. Leonneti's lab and adapted by us for cryo-ET sample preparation. In brief, the cells were lysed using a hypotonic homogenization buffer (25 mM Tris-HCl pH 7.5, 50 mM sucrose, 0.2 mM EGTA, 0.5 mM MgCl$_2$), and shearing forces were generated using a 23-G syringe. To protect the organelle membranes, the lysate was immediately mixed with a sucrose buffer (2.5 M sucrose, 0.2 mM EGTA, 0.5 mM MgCl$_2$) to re-equilibrate the homogenates to an isotonic osmolarity. The nuclear fraction was separated by centrifuging the lysate at 1,000$g$ for 10 min. The lysosomes with the LAMP1–GFP tag from this lysate were then purified on-grid using functionalized electron-microscopy grids[48,49]. This technology was adapted for organelles at the CZ Imaging Institute and University of California, San Francisco, in a joint collaboration with D. Agard's lab. The lysosomes were captured on-grid using anti-GFP nanobodies. These anti-GFP nanobodies were attached to the maleimide groups covering the surface of the grid using a kck linker. All lysis steps were performed at 4 °C, and the lysate was kept on ice until plunge freezing.

Plunge freezing was done using a Leica GP2, a Whitman #1 blotting paper and liquid ethane at –180 °C as a cryogen for fast freezing. The functionalized grid was loaded into the chamber, which was set at 4 °C and 95% humidity. The lysate was pipetted up and down (20 strokes) in 10-µl volumes over 10 repeated rounds, resulting in an overall application of 100 µl lysate to the grid surface. The grid surface was then washed twice with PBS by pipetting. Before the purified protein species was added, most of the buffer volume left on the grid was pipetted away, leaving the grid and the attached lysosomes just sufficiently hydrated. This limited the dilution of the added purified protein species. Each of the purified proteins was added one at a time in 1-µl volumes as follows: (1) THG (from bovine thyroid, Sigma-Aldrich T9145) at a concentration of ~17.8 mg ml$^{-1}$ (UV absorbance at 280 nm ($A_{280}$), 19.35), (2) apoferritin (from equine spleen, Sigma-Aldrich 178440) at a concentration of ~5 mg ml$^{-1}$ ($A_{280}$, 4.73), (3) β-galactosidase (from *E. Coli*, Sigma-Aldrich G5635) at a concentration of ~6 mg ml$^{-1}$ ($A_{280}$, 13.79), (4) β-amylase (from sweet potato, Sigma-Aldrich A8781) at a concentration of ~5 mg ml$^{-1}$ ($A_{280}$, 4.75), (5) human serum albumin (HSA) (Sigma-Aldrich A3782) at a concentration of ~50 mg ml$^{-1}$ ($A_{280}$, 20.15), and (6) virus-like particles (from collaborators at NYSBC) at a concentration of ~7.5 mg ml$^{-1}$ ($A_{280}$, 26.91). The target concentration (after the sixfold dilution due to mixing) for all of the purified species (except for HSA) was 5 µM. The deficiencies in hitting the 5-µM target, calculated as percentages ((5 µM − measured concentration) / 5 µM), were: (1) THG 15%, (2) apoferritin 33%, (3) β-galactosidase 27% and (4) β-amylase 16%. The HSA concentration was kept high because it served as a background protein. The VLP concentration could not be elevated further because of difficulties with volume handling at high concentrations. Despite not reaching the target concentration, there were sufficient quantities of each target protein to generate a subnanometer-resolution 3D reconstruction, with the exception of the smallest protein, β-amylase. After all the purified proteins were added to the grid, back-side blotting was done for 6 s, and the grid was plunged into liquid ethane and then stored in liquid nitrogen.

### Data collection

All 1,070 tilt series were collected on one grid on a Krios G4 equipped with an X-FEG electron gun, a Falcon 4i direct electron detector, and the SelectrisX energy filter. The pixel size was set to 1.51 Å per pixel, and the total dose to 62.93 e$^-$ Å$^{-2}$ linearly spread over 31 tilt images spanning a range of –45° to +45° in 3° increments. The software used for data collection was TFS Tomo 5. Utilizing the beam-image shift data collection feature, we imaged nine targets at each stage position. The movie frames were saved in the EER format.

For a rapid quality check of the sample, 2D data were also collected on the same grid for single-particle analysis. The virus-like particles were refined to a resolution of 3.93 Å. In brief, 124 movie frames were collected, and all downstream processing was done in CryoSparc[45]. To generate templates for VLPs, 12 micrographs were manually picked. Template matching resulted in 3,844 particles, which were filtered to 606 particles after 2D classification. Ab initio model generation and homogeneous refinement resulted in a 3.93-Å map of an icosahedral VLP.

### Data processing

Motion correction, tilt-series alignment and tomogram reconstruction were all performed using AreTomo3 v1.0.7 (ref. 34). Specifically, raw frames were partitioned into non-overlapping groups of 2,000 frames each, and batches of 10 frames in each group were integrated to generate rendered frames. Every two and four rendered frames were summed and used to measure global and local motion, respectively. Motions measured on group sums were then interpolated to individual rendered frames for more accurate correction of rapid motion. For local motion estimates, these integrated frames were further subdivided into 5×5 patches. Both gain- and motion-corrected frames were summed to generate corrected tilt images. CTF parameters were estimated for each tilt image. These tilt images were then aligned into tilt-series, with global alignment followed by 4×4 patch-based local alignment. Tomograms were reconstructed using weighted-back projection, either with (for denoising) or without (for 3D template matching) applying a local CTF deconvolution and correction to the tilt-series (Supplementary Table 3). Even and odd pairs of CTF-deconvolved tomograms were produced to train a denoising model using DenoisET[34], an in-house implementation of Noise2Noise[50]. This denoising model was trained for 10 epochs on subvolumes extracted from 43 pairs of even and odd CTF-deconvolved tomograms before being applied to full tomograms to denoise the full dataset. Annotation of the ground-truth labels was performed on these tomograms, which had the incorrect hand. The tilt-axis orientation and tomogram handedness were subsequently corrected by reconstructing tomograms with AreTomo3 v2.0.10 while maintaining the original global and local alignments. The ground-truth annotations were also mapped to these correct-handed tomograms, and all data released on the Portal have the correct hand.

### Generating ground truth

Below, we first briefly outline the in-house tools that were developed as part of these efforts (Supplementary Table 1) and then describe how these were implemented into specific workflows for each particle of interest. All of these tools are open-source and available on Github.

DenoisET[34] implements the Noise2Noise algorithm[50] for denoising cryo-ET tomograms. This ML algorithm learns to denoise imaging data from training on paired noisy measurements and is thus suitable for methods, such as cryo-ET, in which the clean signal is difficult to realistically simulate and cannot be measured. However, because cryo-ET data are collected as a series of frames of the same underlying field of view but with different realizations of the stochastic noise process, training data can be generated by reconstructing paired tomograms after splitting the acquired frames into half-sets. Our implementation relies on a similar U-Net architecture as that used in Topaz-Denoise[51] and leverages the CTF-deconvolved tomograms produced by AreTomo3 (ref. 34) to improve contrast enhancement. The increased SNR provided by denoising proved critical for our manual annotation efforts and benefited our machine-learning labeling workflows, which performed better on denoised projection data than on weighted-back projection data.

Copick[41] is a storage-agnostic and server-less platform designed specifically for cryo-ET datasets. This package permits efficient access

of tomograms and annotations both programmatically, to assist algorithm development, and visually in ChimeraX[52] and napari[37], for inspection and manual labeling. Specifically, tomograms are stored in the multi-scale OME-Zarr[38] format to enable rapid and parallel data loading from any file system at different resolutions. Annotations are associated with a particle class, and other unique identifiers and can be overlaid in an editable format on static copies of reference labels. Collectively, these design choices enabled a large team to manually curate our initial picks in parallel and simplified methods development by providing a unifying framework for data access and storage. We leveraged this framework to easily transfer annotations among all of the in-house tools described here.

Slab-picking was developed as an alternative to 3D template matching to generate initial candidate picks. In this approach, tomograms were divided along the $z$ axis into slabs of uniform thickness and projected along that axis to increase the SNR. These projections were then uploaded into CryoSPARC[45] as mock micrographs. Conventional 2D approaches for particle picking, such as blob-picking and template matching, were applied to locate candidate particles in each micrograph, and 2D classification was performed to remove false positives by manually rejecting classes judged to be incorrect. The locations were then mapped back to their corresponding tomograms, and the $z$ height of each particle was refined on the basis of local intensity statistics. Owing to time constraints, we were limited to generating candidate picks for all particle types from non-overlapping 300-Å-thick slabs. However, we expect that more hits could be obtained by using a sliding-window approach to retain particles positioned at the slab boundaries and adjusting the slab height to match the particle size.

ArtiaX[53] is a ChimeraX[52] plug-in that provides a toolbox for visualizing, selecting and editing particle picks in tomograms. This package was extended in several ways to facilitate and accelerate large-scale manual curation efforts. First, the package was rendered interoperable with the copick framework to enable annotators to rapidly render tomograms at different locations and at distinct voxel spacings, quickly switch between tomograms of interest and independently curate the same set of candidate picks in parallel. Second, shortcut keys were added to scan candidate picks by recentering the field of view on each particle, enabling fast deletion of false positives. Finally, a new feature provides orthoslice views through the tomogram to aid identification and disambiguate particles that are indistinguishable in projection.

Copicklive provides an interactive web viewer to track picking statistics and tomogram curation in real-time. These features proved useful for maximizing annotation coverage across the full set of tomograms and reducing duplicate efforts during a week-long manual picking marathon involving 42 participants who collectively annotated 147 tomograms and more than 29,000 particle annotations. In addition, the interface provides tools to facilitate particle rejection and class reassignment on the basis of 2D projections of subvolumes centered around candidate picks. Because copick is the foundation of this software package, the results of manual curation are automatically saved in a standardized format that can be easily accessed by other software.

DeepFindET is an adaptation of the DeepFinder[10] package, a convolutional neural network (CNN)-based algorithm to simultaneously label multiple particle types in cellular cryo-ET data. The model architecture follows a U-Net design, which is an encoder–decoder network originally developed for biomedical image segmentation. Although initial results from the standard DeepFinder package were promising, the following extensions significantly improved performance on our phantom dataset. First, we added residual connections in the convolutional layers to enhance gradient flow and improve training stability, reducing the risk of overfitting. Second, additional intensity-based augmentations (for example, random scaling of the histogram or distorting the tomogram with random additive Gaussian noise) were added to diversify the training data. These augmentations reduce the risk of overfitting and ensure that the network can generalize for challenging low-SNR regions. Third, we replaced the original semi-supervised clustering approach used for extracting 3D coordinates with a size-based selection scheme, in which the cut-off was set to two-thirds of each particle's maximum dimension to mitigate extraction bias and increase confidence in particle labels during inference. The model was trained on three sets of picks: (1) 24 synthetic tomograms containing a mixture of target particles, (2) five manually annotated tomograms in the phantom dataset and (3) curated picks from the initial round of template matching and slab picking.

CellCanvas is a flexible tool for building geometric models of cellular architecture, with an associated napari[37] plug-in to enable interactive painting-based segmentation and model refinement. CellCanvas's segmentation abilities were used to generate an intentionally over-picked set of candidate particles to minimize false negatives. The first step of this pipeline generated a voxel-wise embedding for each tomogram in the phantom dataset using a Swin UNETR model, which was pre-trained on medical imaging data[54] and fine-tuned on four synthetic tomograms generated using PolNet[35]. Clusters in embedding space were transformed into segmentation masks by interactively training an XGBoost classifier on manual annotations. These segmentation masks were grouped by the Watershed algorithm into similar regions, and the centroid of each region was labeled as the particle with the highest local probability density. Although we found the combination of pre-computed embeddings and quick-to-train interactive models highly effective for labeling the phantom data, CellCanvas was intentionally built with a plug-and-play design that enables replacing the embedding model and classifier with other models for increased flexibility.

Minislab curation was developed to facilitate particle curation. In this approach, subvolumes centered on individual particles were extracted from the tomograms, and the intensity was integrated along the $z$-axis to generate per-particle 2D projections. These 'minislabs' were then tiled to generate mock micrographs for further processing in CryoSPARC[45]. The particles were cleaned up and curated through various combinations of manual picking, 2D classification, running ab initio reconstruction with multiple classes and applying Topaz[12], a CNN-based particle picker that was retrained for each particle class. For VLPs, the 80S ribosome and apoferritin, we manually annotated 2, 14 and 19 mock micrographs (each containing 240 particles), respectively, per class and used these labels to train a Topaz model for each particle. The trained Topaz models were then applied to the remaining mock micrographs. To increase the accuracy of our labels, we inspected minislabs of both the rejected particles to rescue any true positives and the accepted particles to eliminate any false positives during a final round of manual curation. The selected particle projections were mapped back to their positions in the tomograms. Although minislab curation cannot distinguish between particles that appear similar in projection, curation of 2D projections was more efficient than that for 3D subvolumes and also benefited from the increased SNR.

Synthetic tomograms were generated using PolNet[35], a Python package designed to simulate cryo-ET data to train deep learning models. PolNet simulated cellular components, including membranes with double-layer Gaussian profiles, as well as randomly oriented macromolecules derived from PDB[55] files, which were distributed randomly in a 180-nm-thick tomogram. After constructing the 3D tomogram density map, a tilt series was generated by producing perfect projection images, which were then translationally misaligned and corrupted with random Gaussian additive noise to mimic experimental distortions. Finally, the synthetic tomograms were reconstructed using IMOD[56].

The tools described above and other published software packages were stitched into several workflows to generate ground-truth labels (Fig. 1 and Extended Data Fig. 7). Owing to the complexity of the curation workflow, we first describe the overall workflow, and then the

specific approach used for each species. We generated enough initial picks that could be used to train ML models, and to subsequently clean those picks through 2D and 3D classification supervised by human experts. Initial picks were generated either using PyTom[43,44] 3D template matching or 2D slab-picking. These picks contained false positives, which were filtered out using 2D classification, Topaz[12] and manual cleaning. However, there were not enough remaining picks for model training, and additional manual picking was performed by human experts, especially for THG and β-galactosidase. Manual picking was done using ArtiaX[53] and ChimeraX[52]-copick plugin. These picks were used to train DeepFindET and the CellCanvas classifier, both in a multi-class approach. Picks from PolNet[35]-simulated tomograms were also used in the training. Pre-training these models on synthetic data before fine-tuning on experimental tomograms improved prediction accuracy by allowing the networks to first learn structural features in a noise-controlled environment. However, the best results were obtained after including picks from the experimental phantom dataset. The picks from DeepFindET were used for downstream curation for all the species. CellCanvas picks were retained only for THG and β-galactosidase because there were too many ribosome, VLP and apoferritin picks to process effectively.

DeepFindET and CellCanvas were used to generate thousands of picks per species that were filtered out using 2D and 3D classification, Topaz[12], manual cleaning and a final step of resolving duplicate labels between particle classes. Through multiple iterations of this process, the human experts developed a familiarity with the shape of the particles in the dataset, enabling them to manually curate the final picks after working on this problem for several months. In what follows, we describe the specifics of the workflow for each annotation target (Extended Data Fig. 7). We highlight the workflow for the 80S ribosomes in more detail (Extended Data Fig. 8). The workflows and tools used for the other species are similar to those used for ribosomes.

**Ribosomes.** Ribosomes were initially picked using PyTom 3D template matching[43,44] on weighted backprojection tomograms (10 Å per pixel). Templates were generated internally in PyTom from published maps[28] (EMDB ID: 3883). Another set of picks was generated using 2D slab picking in parallel (template matching in CryoSPARC[45] using micrographs generated from 30-nm slabs of CTF-corrected and denoised tomograms with a pixel size of 5 Å per pixel). The PyTom and CryoSPARC picks were merged, and duplicates were removed. This merged set was cleaned using 2D classification, and Topaz was applied on minislabs. DeepFindET was trained on these picks (and picks of other species simultaneously in a multi-class approach), which resulted in tens of thousands of new picks and yielded 50–100 ribosomes per tomogram. After this initial set of picks was generated, we started the process of curation, which involved removing all the false positives, eliminating off-centered picks and finding all full 80S ribosomes (Extended Data Fig. 8). The curation was done in CryoSPARC in this order: (1) a Topaz[12] model pre-trained on the same data was applied, (2) the selected Topaz picks were manually cleaned and false negatives (ribosomes that were incorrectly rejected by Topaz) were rescued, (3) 2D classification was performed and good classes were selected, (4) ab initio refinement was carried out with two classes, (5) the 80S ribosome class was homogeneously refined and (6) another round of 2D classification was carried out and good classes were selected. This set of 80S ribosome particles was exported from CryoSPARC and inspected for duplicate labeling with other particle species. Finally, these picks were used in a Relion-5 (ref. 57) refinement workflow without any classification that resulted in a 3.6 Å map based on the global FSC resolution.

**Virus-like particles.** Steps 1–3 were repeated for VLPs (template used for template matching: EMDB ID-41917 (ref. 23). This resulted in the final set of VLPs, including mostly icosahedral particles along with other geometries that were less frequent, such as tubular VLPs.

**Apoferritin.** The initial picks for apoferritin were generated using the slab method on denoised tomograms (5 Å per pixel), which divided each tomogram into 30-nm slabs. Each slab was then projected along the z-axis and processed as a 2D micrograph. To generate the first set of apoferritin picks, 2D template matching in CryoSPARC[45] was used. These picks were cleaned using 2D classification and Topaz[12]. Candidate particles were mapped back to their tomograms, and the depth of each particle was refined by locating the z-coordinate with the maximum integrated intensity in a subvolume centered on the particle. The refined picks were then combined with manual picks to create a training set for DeepFindET. The trained model generated ~40,000 picks, which were cleaned again (using 2D classification, Topaz and manual curation in CryoSparc), resulting in 20,000 particles. Of the 20,000 particles that were filtered out, a fraction of true positives were not captured, because the apoferritin particles in this sample formed clusters (like beads on a string) and there were incomplete individual shells, which together introduced errors in centering the picks. Generating minislabs from these off-centered picks adversely affected the contrast and made it difficult to identify and average them.

**THG, β-galactosidade, β-amylase.** The initial picks for these three most challenging species were also generated using the slab method, similar to apoferritin. However, this yielded only a few hundred particles for THG and β-galactosidase after 2D classification, and no particles for β-amylase. The 2D classes also suffered significantly from preferred orientation caused by underpicking. Therefore, five tomograms were manually annotated by human experts to capture more views of THG and β-galactosidase. β-amylase was very difficult to pick and remained underpicked. These picks were used to train DeepFindET and the CellCanvas classifier. Both models generated several thousand picks for THG and β-galactosidase. The results of each model were processed separately and eventually merged for each species. These picks were predominantly cleaned by 2D classification. The results showed that the preferred-orientation problem was resolved. Through the 2D classification of THG and β-galactosidase picks, some 2D classes of β-amylase emerged. Duplicates were removed on the basis of a distance threshold of 185 Å and 125 Å for THG and β-galactosidase, respectively. Unique picks from the two models were combined. The merged sets were manually curated, followed by ab initio reconstruction with two classes. Particles that contributed to the correct reconstruction class were retained.

Finally, for all six species, we generated minislabs from any positions that belonged to more than one particle class and manually assigned one label based on visual inspection to remove these inter-class duplicates. The final particle lists from these curation efforts were considered the ground truth annotations.

### Relion-5 subtomogram averaging

After finalizing the coordinates for all six species of protein complexes, we used a Relion-5 (ref. 57) subtomogram averaging pipeline to reconstruct subnanometer-resolution (or close to subnanometer-resolution) 3D maps. To streamline the processing steps for all species, we developed a semi-automated pipeliner called pyRelion.

PyRelion handles the importing of coordinates and tilt series processed by AreTomo3 into a format suitable for Relion-5. With the imported data, we obtained initial orientations by running 3D classification at bin 6 (9.06 Å per pixel) with a single class against a published reference. The particles were then refined at that resolution using refine3D before the particles were extracted and reconstructed at bin 2 (3.02 Å per pixel). A subsequent refinement was performed at bin 2 using the reconstructed map as the reference and a particle-shaped mask.

Once the refinements reached close to the bin 2 Nyquist resolution (~6.04 Å)—which was the case for the ribosomes, VLP and apoferritin—subtomograms were re-extracted at the unbinned pixel size of 1.51 Å per pixel. Following an initial round of unbinned 3D refinements,

these datasets underwent five cycles of CTF refinement and Bayesian polishing. Each cycle includes the following operations in Relion-5: (1) 3D reconstruction, (2) CTF refinement to improve per-particle defocus, (3) Bayesian polishing for per-particle motion correction, (4) extraction of subtomograms with new shifts and orientations, (5) 3D reconstruction of newly extracted particles and (6) 3D refinements for further resolution improvement. In all cases, refinements and resolution estimates were performed using the same soft-edge mask. The masks were calculated from the first initial 3D reconstruction following unbinned subtomogram extraction, which was low-pass filtered to 10-Å resolution, set to an intensity binarization thresholded at 1 standard deviation for the map values, and a soft-edge padding of 10 pixels. The final FSCs are reported in Extended Data Fig. 9.

### Reporting summary

Further information on research design is available in the Nature Portfolio Reporting Summary linked to this article.

### Data availability

The full phantom dataset is available on the CryoET Data Portal under deposition ID CZCDP-10310. This deposition is split among a training set (7 tomograms, DS-10440); a public test set (121 tomograms, DS-10445); and a private test (364 tomograms, DS-10446) based on the subsets used for the associated competition on Kaggle (https://cryoetdataportal.czscience.com/competition). The deposition also includes the annotations from that competition and a synthetic dataset (27 tomograms, DS-10441) designed to mimic the phantom dataset.

### Code availability

Software packages used in this study for tomogram reconstruction, annotation generation and annotation curation are released under open-source licenses and available in public repositories. Supplementary Table 1 provides a summary of packages and locations. The source code for pyRelion will be released soon and is available on request.

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

## Acknowledgements

We thank E. Lundberg (Stanford), E. Zhong (Princeton), T. Wagner (Max Planck Institute of Molecular Physiology), T. Bepler (New York Structural Biology Center (NYSBC)), R. Kiewisz (NYSBC), A. Burt (Genentech) and L. Gaifas (Institut de Biologie Structurale, Grenoble), who provided valuable insights for the design of the challenge. We thank M. Leonetti, S. Vaid, M. Vangipuram and R. Baltazar (CZ Biohub SF) for generating the HEK293T LAMP1–GFP cell lines and sharing their cell lysis protocol; F. Wang and S. Sander (University of California, San Francisco (UCSF)) for providing their grid functionalization protocol, grids and help with optimizing the protocol further for organelles; P. Jin (UCSF) for providing and modifying the anti-GFP–nanobody construct; and M. Kopylov and C. Dubbledam (NYSBC) for providing VLPs. We thank the following individuals from the Chan Zuckerberg Initiative (CZI) for participating in the pickathon: A. Anderson, B. Nelson, J. Ni, E. Chou, J. Gadling, K. Khandwala, C. Chiu, A. Jones, T. Huang, J. Pourroy, D. McCarthy, A. Sweet, E. Wang, K. Ewing, M. Caton, M. Venkatakrishnan and K. Evans. The following CZII members participated in the pickathon: Y. Cho, N. Borja, N. Hill, C. Villegas, S.-H. Sheu, G. Margulis and N. P. Soldan. We acknowledge the following individuals from CZI SciTech for contributing to the development of the the CryoET Data Portal: J. X. Ni, J. Gadling, M. Venkatakrishnan, K. Evans, J. Asuncion, A. Sweet, J. Pourroy, Z. S. Wang, K. Khandwala, B. Nelson, D. McCarthy, E. M. Wang, R. Agarwal, T. Smith, B. Chu, D. Sadgat, E. Hoops and J. Larsen. We thank K. Maitland and S. Otte for their support in the planning and execution of the Kaggle competition associated with this dataset, S. Yammine provided valuable feedback on the manuscript text. The CZ Imaging Institute is fully funded by the Chan Zuckerberg Initiative (CZII-2023–327779). Some schematic elements in Extended Data Fig. 1a,b were made using BioRender (Paraan, R. & Serwas, D. BioRender.com/r59u258 (2024)).

## Author contributions

These authors contributed equally to this work and were listed alphabetically: A.C., U.H.E., J.H., S.K., D.K., E.A.M., D.S., H.S., F.W., Z.Z. and S.Z. M.P. and B.C. developed the idea for a phantom sample and managed the project. M.P., H.S., D.S. and F.W. optimized grid functionalization and freezing of organelles and carried out the sample preparation. M.P. and E.A.M. carried out the data collection. M.P. curated the tomograms on the basis of sample quality and tilt-series alignment quality. S.Z., A.P., Y.Y., J.S. and M.P. developed the data-processing pipeline. S.Z. and A.P. developed AreTomo3 CTF deconvolution and DenoisET. S.Z., A.P. and Y.Y. developed the slab method. U.H.E. developed copick and ChimeraX-copick plugin. K.H. and Z.Z. developed CellCanvas and copicklive. J.S. developed DeepFindET. D.K. and J.S. developed the 3D refinement pipelines. M.P. and Y.Y. manually picked initial training sets. M.P., A.P., Y.Y. and J.S. prepared the final curations for all the picks for all the species. J.S. and J.H. carried out the high-resolution 3D refinements. S.K., J.S., K.H. and Z.Z. prepared the example notebooks. U.H.E. and A.C. developed the CryoET Data Portal. M.P., A.P., U.H.E., K.H., Z.Z. and B.C. wrote the manuscript. All authors reviewed the manuscript. M.H., D.A.A., C.S.P. and B.C. provide overall leadership for projects at the Chan Zuckerberg Imaging Institute.

## Competing interests

The authors declare no competing interests.

## Additional information

**Extended data** is available for this paper at

**Supplementary information** The online version
contains supplementary material available at

**Correspondence and requests for materials** should be addressed to
Kyle Harrington or Mohammadreza Paraan.

**Peer review information** *Nature Methods* thanks Randy Read and the
other, anonymous, reviewer(s) for their contribution to the peer review
of this work. Peer reviewer reports are available. Primary Handling

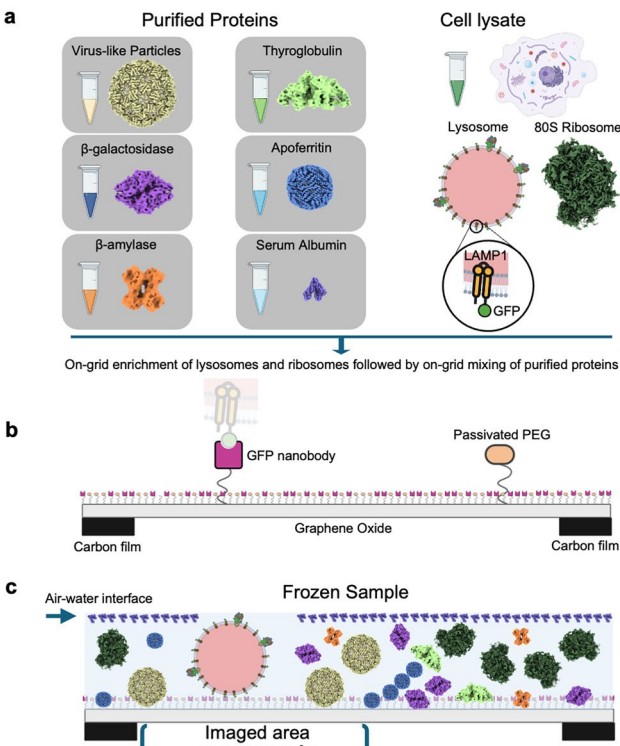

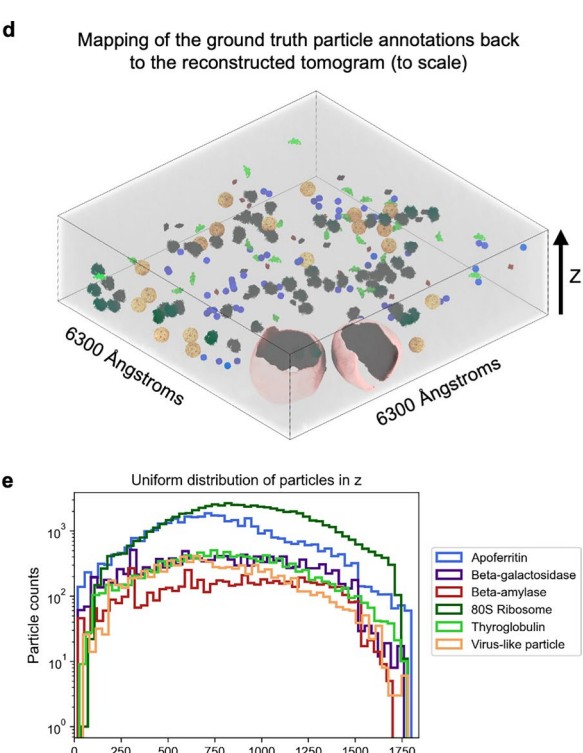

**Extended Data Fig. 1 | The composition of the phantom sample. a**. Two sources were used in the preparation of the phantom sample: purified proteins and cell lysate. From the cell lysate, lysosomes were captured by affinity interactions between LAMP1-GFP and anti-GFP nanobody. Ribosomes co-purified because of their high abundance. **b**. Cross-section (1.2-μm width) of the functionalized EM grid surface prior to sample application. The carbon film (20-nm height) serves as a support for, from bottom to top: graphene oxide (1-atom thickness), combination of passivated PEG and PEG-maleimide-anti-GFP-nanobody. **c**. A schematic cross section through the sample on the grid; note proteins are not drawn to scale. The proteins are frozen in time in glass-like ice (ice without crystals). Tilt series were collected with a field of view of 6300 Å x 6300 Å in

regions excluding the carbon film. **d**. A 3D view of a typical phantom tomogram. Annotations drawn to scale. All the annotated particles (except for HSA) are placed in their true positions in a tomogram with the boundaries of the 3D volume outlined. The membranes are rendered as surfaces. **e**. The distribution of all the annotated species in Z shows an almost uniform distribution across the thickness of the sample. This highlights the advantage of using the functionalized grids to capture lysosomes, which created a spacer effect that prevented the majority of proteins from adhering to the air-water interface. Elements of panels **a** and **b** were created in BioRender (Paraan, R., Serwas, D. (2024) BioRender.com/r59u258).

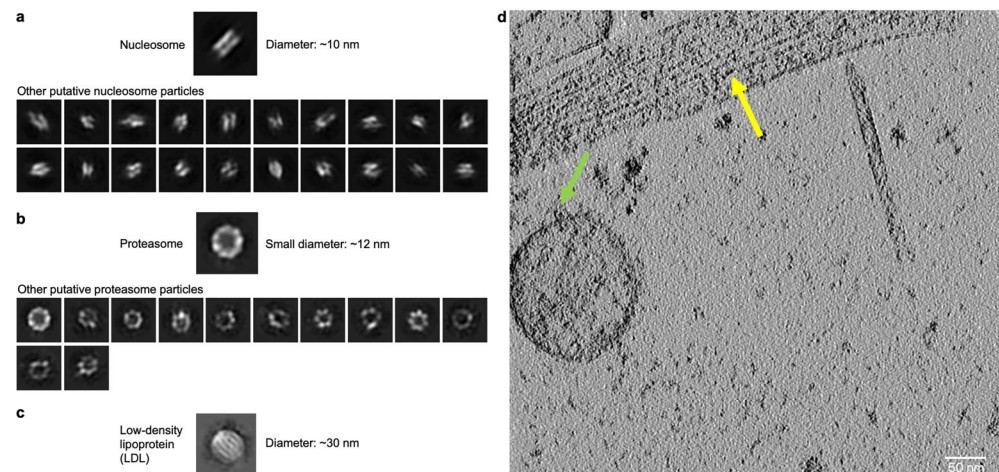

**Extended Data Fig. 2 | Other species of proteins that co-purified from the lysate. a**. 2D classes of nucleosomes (4,776 particles). They are prevalent in the nucleus and co-purify after cell lysis. **b**. 2D classes of proteasomes (565 particles). They are cytosolic and less prevalent than nucleosomes in the dataset. Their small diameter as seen in the 2D class is ~12 nm and their large diameter is ~16 nm. **c**. A 2D class of LDL complexes (~100 particles). They are less prevalent than nucleosomes and proteasomes and attach to membranes. **d**. An example tomogram slice showing a bundle of actin filaments (yellow arrow) and transmembrane protein complexes (lysosomal proteins, green arrow).

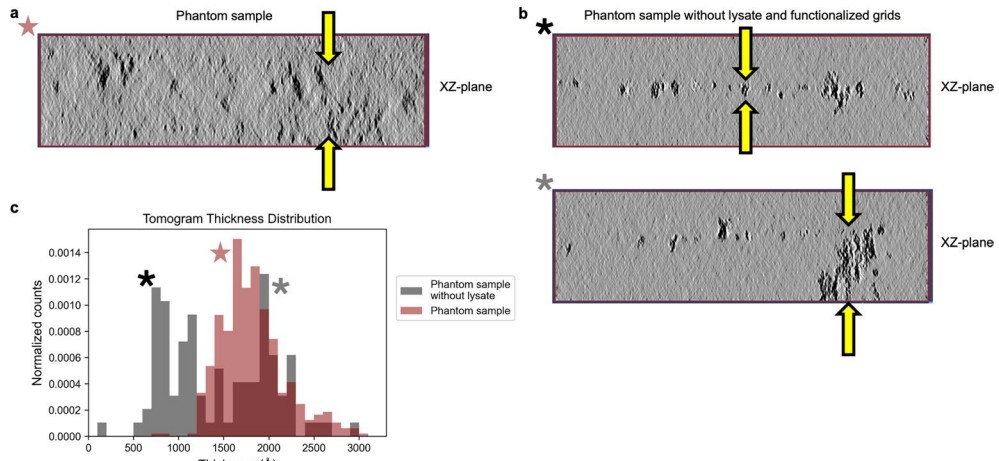

**Extended Data Fig. 3 | Graphene-oxide functionalized grids and cell lysates create more uniform and thicker samples. a**. XZ view of a tomogram from the phantom dataset. Yellow arrows point to the boundaries of the sample and indicate a thicker volume compared to b. This thicker volume captures the typical noise behavior in a cryoET dataset. **b**. XZ views of a sample prepared in-house which is just a mix of the 5 seeded proteins (VLP, apoferritin, HSA, THG, beta-amylase). Without using graphene-oxide functionalized grids and lysates, the sample is mostly one thin layer of proteins (top). This introduces issues like aggregation (bottom) and SNR similar to that observed for single particle analysis as opposed to cryoET. **c**. The thickness of the tomograms from the two datasets were estimated using AreTomo3. The phantom dataset (**a**, star) shows a uniform distribution. The phantom dataset without cell lysates and functionalized grids (**b**) has a bimodal distribution: one peak (around 600 Å) for tomograms that are one particle-layer thick (**b**, top, black asterisk), and another peak (around 2000 Å) for tomograms with large aggregations (**b**, bottom, grey asterisk).

a    Clusters of small densities at the air-water interface

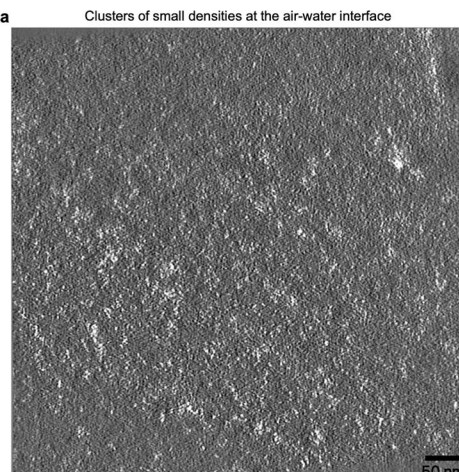

b    Orthogonal views of the air-water interface (yellow arrow)

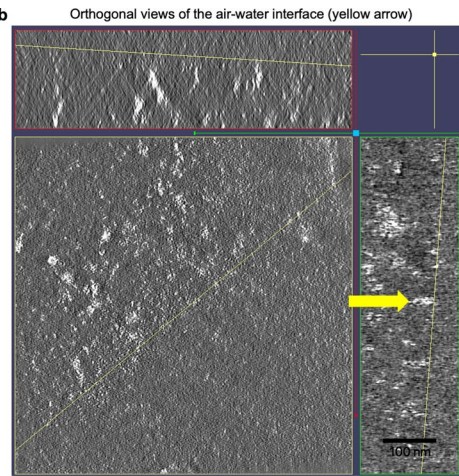

**Extended Data Fig. 4 | The air-water interface in the phantom dataset. a**. A slice through the air-water interface showing a large population of small densities as white speckles. Scale bar is 50 nm. **b**. The same tomogram from a, seen in orthogonal views. The yellow line indicates the common line between the tilted plane that generated panel a and each orthogonal view. The yellow arrow shows a larger density at the air-water interface. The contrast was inverted to white particles and dark background for better visualization of the small densities. Scale bar is 100 nm.

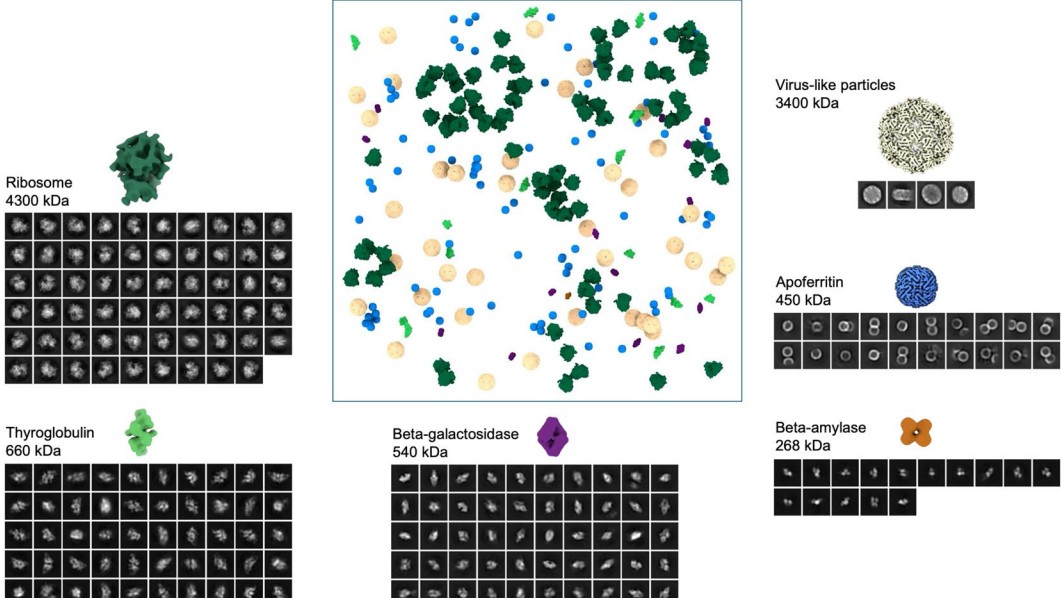

**Extended Data Fig. 5 | Lack of preferred orientation in the phantom dataset.** 2D classes were generated using the minislab method on denoised tomograms with a pixel size of 5 Å/pix. For ribosome, THG, beta-galactosidase, VLP, and beta-amylase species, the 3D maps were generated by performing ab initio reconstruction from these minislabs in CryoSparc. For apoferritin, the 2D classes are generated using the slab method, but the 3D map is from Relion-5. VLP and apoferritin are highly symmetric; therefore, the 2D classes are representative of heterogeneity rather than the orientation distribution.

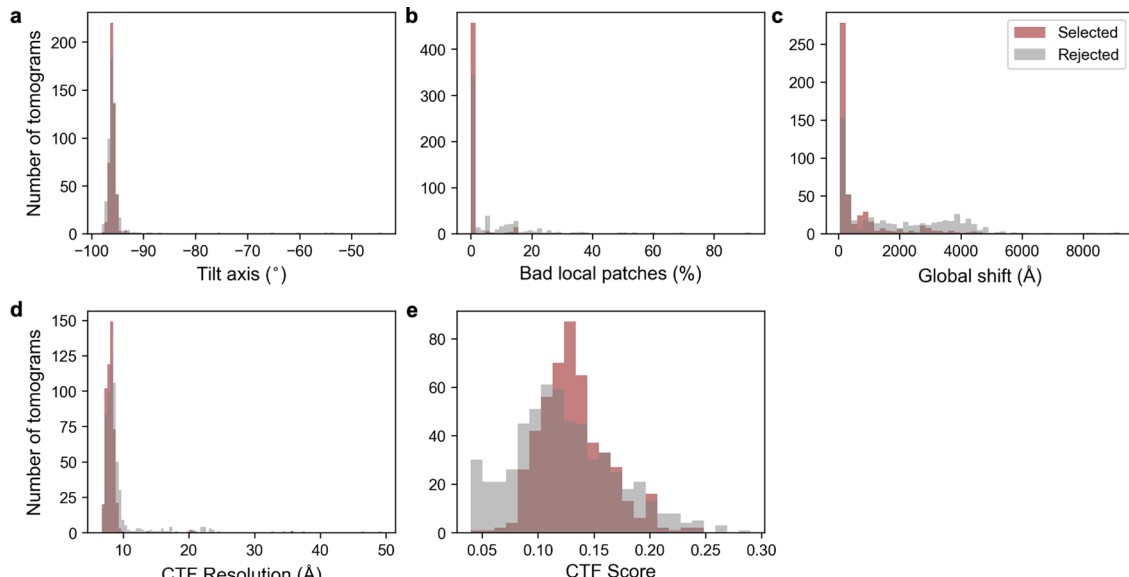

**Extended Data Fig. 6 | AreTomo3 tilt series quality metrics for the phantom dataset (1,070 tomograms total) plotted against the tomograms that were manually curated for quality (492 tomograms selected).** Selected tomograms have the following characteristics: **a**. correct tilt axis angle. **b**. the lowest percentage of bad patches in patch-based tilt series alignment. **c**. the lowest global shifts during data collection. **d**. CTF resolutions are below 10 Å. **e**. high CTF confidence scores.

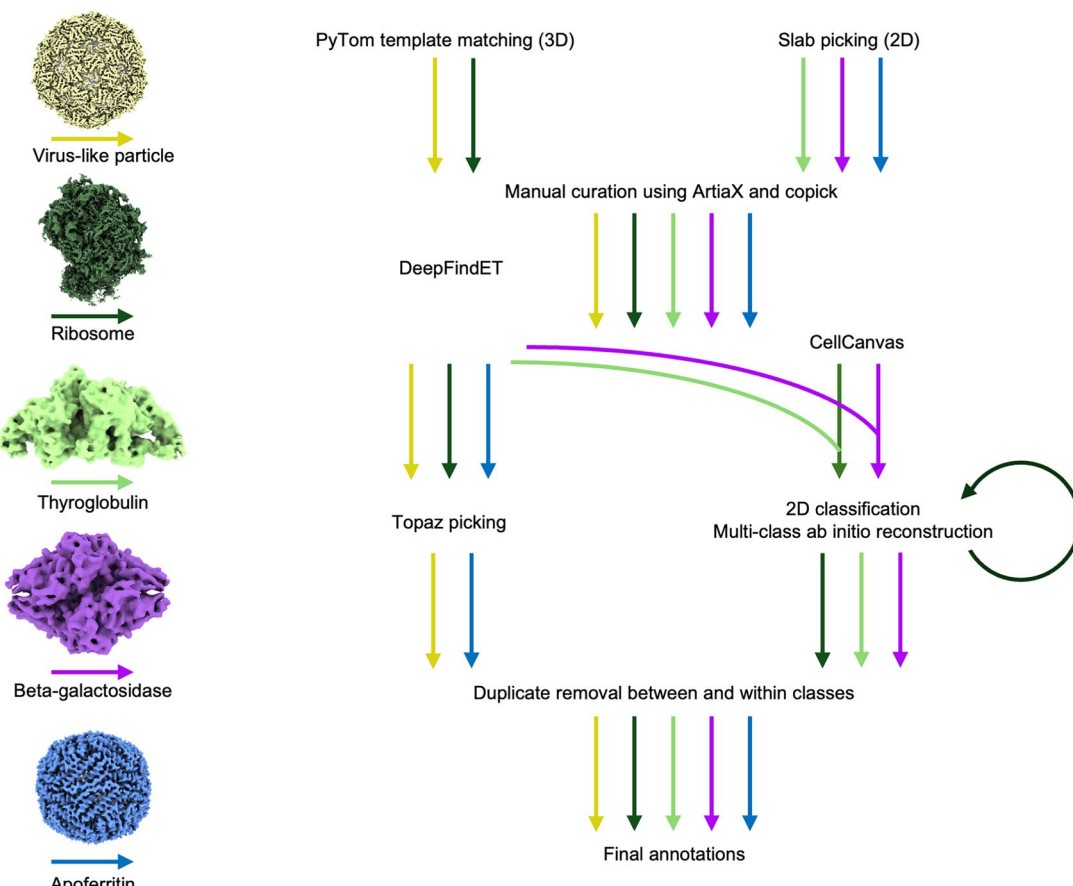

**Extended Data Fig. 7 | Workflow for generating ground truth labels.** Despite applying each of the listed picking and refinement methods to every particle, we found that no single workflow worked for all particles. This schematic traces which tools were applied to generate the final annotations that we refer to as ground truth labels for each particle class, with each particle indicated by the color shown in the legend at the left.

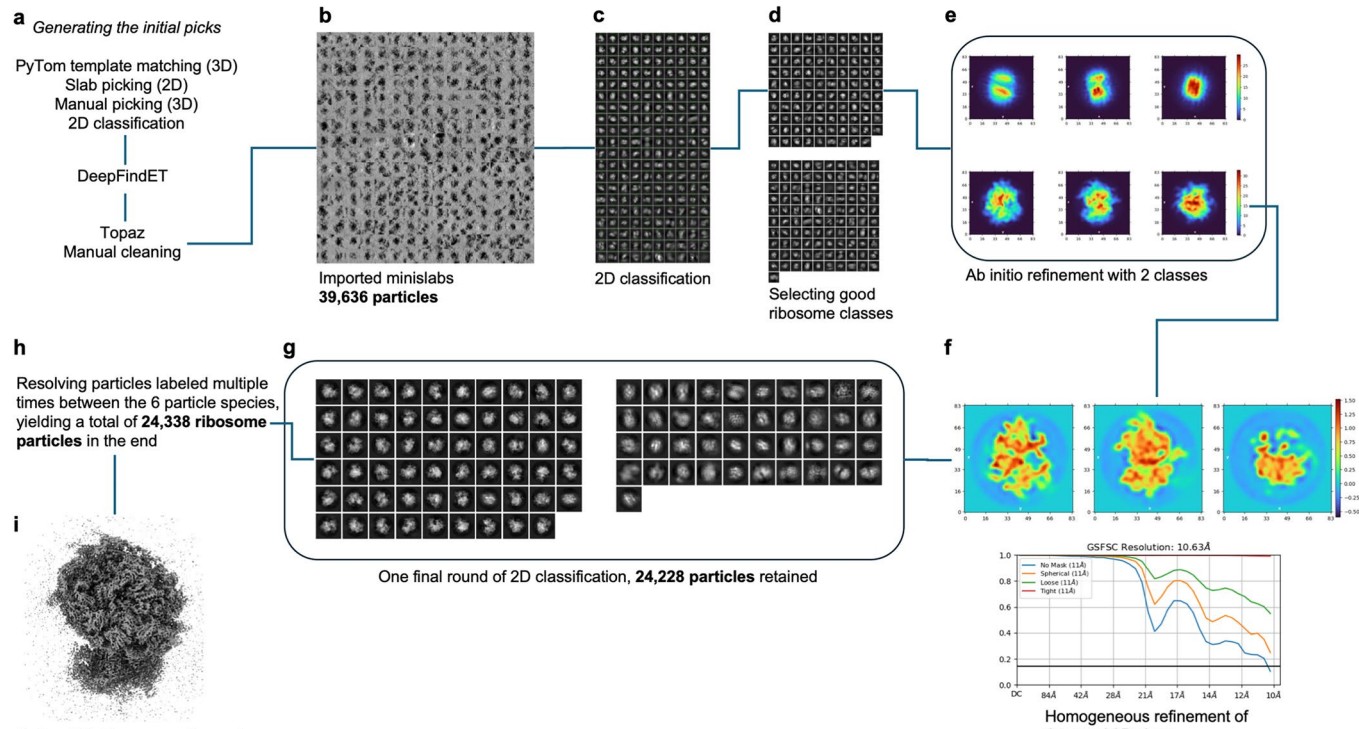

**Extended Data Fig. 8 | Overview of the 80S ribosome curation workflow. a.** An initial set of picks is generated as described in Fig. 1 and the Methods section. **b–g.** These steps are the curation steps where false positives and off-centered picks are removed, and only full 80S ribosome particles are retained. **b.** One mock micrograph containing 240 minislabs of the initial ribosome picks. The minislabs were projected from subtomograms that were CTF-corrected and denoised with a pixel size of 5 Å/pixel. A total of 34,636 picks are imported into CryoSPARC. **c.** 2D classification of minislabs with 200 classes. About 29,000 of all the particles had a "probability of best class" of 0.99, indicating reliable class assignment. **d.** Out of the 200 classes, 99 classes were selected as "good" 80S classes. That is 29,638 particles. **e.** Ab initio refinement was done on the good

classes from d using two 3D classes. Class 1 had a non-sense map, while class 2 was a map of 80S ribosomes with 26,371 particles. **f.** A homogenous refinement of the 80S class converged to the Nyquist limit of the minislabs (10 Å). **g.** To remove any bad particles that could have ended up in the 80S class, another round of 2D classification was done with 100 classes. About 21,000 of the particles had a "probability of best class" of 0.99. Out of 100 classes, 59 were retained with 24,228 particles. These particles were exported from CryoSPARC. **h.** The final curated set was created by adding back 80S ribosome particles that were mislabeled as other species (such as VLPs), resulting in a total of 24,338 particles. **i.** The final highly curated set of 80S ribosome particles were refined in Relion-5 to a global FSC resolution of 3.6 Å (pixel size of 1.51 Å/pix, full resolution without upsampling).

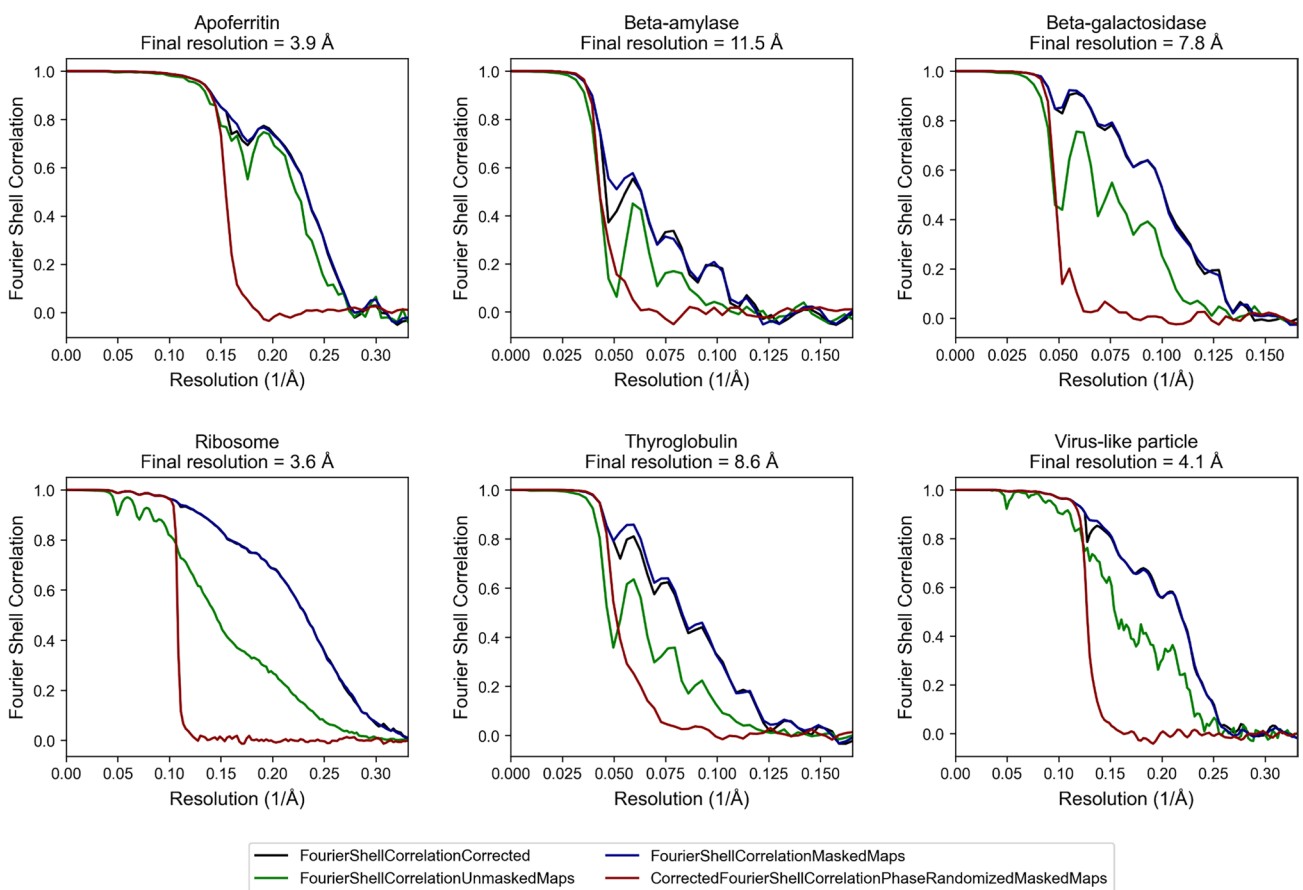

**Extended Data Fig. 9 | Half-map FSC curves from Relion-5 subtomogram averaging. Ribosome** (4.3 MDa): 3.6 Å, 24,338 particles. **VLP** (3.4 MDa): 4.1 Å, 3022 particles. **THG** (660 kDa): 8.6 Å, 6,211 particles. **Beta-galactosidase** (540 kDa): 7.8 Å, 3,113 particles. **Apoferritin** (450 kDa): 3.9 Å, 23,393 particles. **Beta-amylase** (268 kDa): 11.5 Å, 2,464 particles.

# Reporting Summary

## Statistics

For all statistical analyses, confirm that the following items are present in the figure legend, table legend, main text, or Methods section.

| n/a | Confirmed | |
|---|---|---|
| ☒ | ☐ | The exact sample size (*n*) for each experimental group/condition, given as a discrete number and unit of measurement |
| ☒ | ☐ | A statement on whether measurements were taken from distinct samples or whether the same sample was measured repeatedly |
| ☒ | ☐ | The statistical test(s) used AND whether they are one- or two-sided<br>*Only common tests should be described solely by name; describe more complex techniques in the Methods section.* |
| ☒ | ☐ | A description of all covariates tested |
| ☒ | ☐ | A description of any assumptions or corrections, such as tests of normality and adjustment for multiple comparisons |
| ☒ | ☐ | A full description of the statistical parameters including central tendency (e.g. means) or other basic estimates (e.g. regression coefficient) AND variation (e.g. standard deviation) or associated estimates of uncertainty (e.g. confidence intervals) |
| ☒ | ☐ | For null hypothesis testing, the test statistic (e.g. *F*, *t*, *r*) with confidence intervals, effect sizes, degrees of freedom and *P* value noted<br>*Give P values as exact values whenever suitable.* |
| ☒ | ☐ | For Bayesian analysis, information on the choice of priors and Markov chain Monte Carlo settings |
| ☒ | ☐ | For hierarchical and complex designs, identification of the appropriate level for tests and full reporting of outcomes |
| ☒ | ☐ | Estimates of effect sizes (e.g. Cohen's *d*, Pearson's *r*), indicating how they were calculated |

*Our web collection on statistics for biologists contains articles on many of the points above.*

## Software and code

Policy information about availability of computer code

| Data collection | Tilt series were collected by Thermo Fisher Scientific (TFS) Tomo 5 5.1.2. |
|---|---|
| Data analysis | AreTomo3 1.0.7, 2.0.10; DenoisET 0.1.0; ChimeraX-Copick 0.5.0; pytom-match-pick 0.6.0; slabpick 0.1.0; DeepFindET 0.4.0; Topaz 0.2.5; cryoSPARC 4.4.1; CellCanvas (commit f3fa5b6 of https://github.com/cellcanvas/album-catalog); copick-live (beta release); Relion-5 (beta-3 release); PyRelion (beta release); PolNet (Github commit 73cff52, software is not versioned); ArtiaX 0.4-0.4.10 |

For manuscripts utilizing custom algorithms or software that are central to the research but not yet described in published literature, software must be made available to editors and reviewers. We strongly encourage code deposition in a community repository (e.g. GitHub). See the Nature Portfolio guidelines for submitting code & software for further information.

## Data

Policy information about availability of data

All manuscripts must include a data availability statement. This statement should provide the following information, where applicable:
- Accession codes, unique identifiers, or web links for publicly available datasets
- A description of any restrictions on data availability
- For clinical datasets or third party data, please ensure that the statement adheres to our policy

The experimental phantom dataset and associated synthetic dataset are available on the CryoET Data Portal under deposition ID CZCDP-10310 (https://cryoetdataportal.czscience.com/depositions/10310).

## Research involving human participants, their data, or biological material

Policy information about studies with human participants or human data. See also policy information about sex, gender (identity/presentation), and sexual orientation and race, ethnicity and racism.

| | |
|---|---|
| Reporting on sex and gender | N/A |
| Reporting on race, ethnicity, or other socially relevant groupings | N/A |
| Population characteristics | N/A |
| Recruitment | N/A |
| Ethics oversight | N/A |

Note that full information on the approval of the study protocol must also be provided in the manuscript.

# Field-specific reporting

Please select the one below that is the best fit for your research. If you are not sure, read the appropriate sections before making your selection.

☒ Life sciences  ☐ Behavioural & social sciences  ☐ Ecological, evolutionary & environmental sciences

For a reference copy of the document with all sections, see nature.com/documents/nr-reporting-summary-flat.pdf

# Life sciences study design

All studies must disclose on these points even when the disclosure is negative.

| | |
|---|---|
| Sample size | N/A |
| Data exclusions | N/A |
| Replication | N/A |
| Randomization | N/A |
| Blinding | N/A |

# Reporting for specific materials, systems and methods

We require information from authors about some types of materials, experimental systems and methods used in many studies. Here, indicate whether each material, system or method listed is relevant to your study. If you are not sure if a list item applies to your research, read the appropriate section before selecting a response.

### Materials & experimental systems

| n/a | Involved in the study |
|---|---|
| ☒ | ☐ Antibodies |
| ☐ | ☒ Eukaryotic cell lines |
| ☒ | ☐ Palaeontology and archaeology |
| ☒ | ☐ Animals and other organisms |
| ☒ | ☐ Clinical data |
| ☒ | ☐ Dual use research of concern |
| ☒ | ☐ Plants |

### Methods

| n/a | Involved in the study |
|---|---|
| ☒ | ☐ ChIP-seq |
| ☒ | ☐ Flow cytometry |
| ☒ | ☐ MRI-based neuroimaging |

## Eukaryotic cell lines

Policy information about cell lines and Sex and Gender in Research

| | |
|---|---|
| Cell line source(s) | HEK293T cell line generated at Chan Zuckerberg Biohub, San Francisco by Manual Leonneti's group with a knock-in GFP-Tag on the C-terminus of the lysosomal house-keeping protein LAMP1 |

| Authentication | Authentication was performed by sequencing. |
| Mycoplasma contamination | Confirmed to be negative. |
| Commonly misidentified lines<br>(See ICLAC register) | No commonly misidentified cell lines were used in this study. |

## Plants

| Seed stocks | *Report on the source of all seed stocks or other plant material used. If applicable, state the seed stock centre and catalogue number. If plant specimens were collected from the field, describe the collection location, date and sampling procedures.* |
| Novel plant genotypes | *Describe the methods by which all novel plant genotypes were produced. This includes those generated by transgenic approaches, gene editing, chemical/radiation-based mutagenesis and hybridization. For transgenic lines, describe the transformation method, the number of independent lines analyzed and the generation upon which experiments were performed. For gene-edited lines, describe the editor used, the endogenous sequence targeted for editing, the targeting guide RNA sequence (if applicable) and how the editor was applied.* |
| Authentication | *Describe any authentication procedures for each seed stock used or novel genotype generated. Describe any experiments used to assess the effect of a mutation and, where applicable, how potential secondary effects (e.g. second site T-DNA insertions, mosiacism, off-target gene editing) were examined.* |

