## [Peer Review File · Nature Methods]

A Realistic Phantom Dataset for Benchmarking CryoET Data Annotation

Corresponding Author: Dr Mohammadreza Paraan

A version of this paper was originally rejected for publication by Nature Methods, however that decision was reconsidered after appeal by the authors.

Version 0:

Decision Letter:

10th Jan 2025

Dear Reza,

Your Resource entitled "Annotating CryoET Volumes: A Machine Learning Challenge" has now been seen by 3 reviewers, whose comments are attached. While they find your work of potential interest, they have raised some serious concerns which in our view are sufficiently important that they preclude publication of the work in Nature Methods, at least in its present form.

As you will see, the reviewers believe that the ML Challenge is a great initiative, but raise similar concerns to our original reservations about this manuscript. We are open to considering a revision, but think that the paper would be better suited for a shorter format (Brief Communication) that focuses on the characterization of the annotated dataset as a resource for the community. We think that the technical comments of Reviewer #3 would be particularly important to address.

Should you decide to address these criticisms we would be willing to look at a revised manuscript (unless, of course, something similar has by then been accepted at Nature Methods or appeared elsewhere). This includes submission or publication of a portion of this work somewhere else. We hope you understand that until we have read the revised paper in its entirety we cannot promise that it will be sent back for peer-review.

If you are not interested in revising your paper, it may be appropriate for another journal in the Nature Portfolio. If you wish to explore the journals and transfer your manuscript please use our manuscript transfer portal. You will not have to re-supply manuscript metadata and files, unless you wish to make modifications. For more information, please see our [manuscript transfer FAQ](http://www.nature.com/authors/author_resources/transfer_manuscripts.html?WT.mc_id=EMI_NPG_1511_AUTHORTRANSF&WT.ec_id=AUTHOR) page.

If you are interested in revising this manuscript for submission to Nature Methods in the future, please contact me to discuss your appeal before making any revisions. Otherwise, we hope that you find the reviewers' comments helpful when preparing your paper for submission elsewhere.

Sincerely yours,
Allison

Allison Doerr, Ph.D.
Chief Editor
Nature Methods

Reviewers' Comments:

Reviewer #1:

Remarks to the Author:

The authors present the organizational framework of a community challenge they have prepared, addressing the need for faster and more reliable ways to annotate tomographic reconstructions from cryo-EM (cryo-ET). Cryo-ET technology is improving rapidly and becoming more widely available, and annotation is key to using it to understand the 3D organization of cells and tissues.

The idea of a challenge for cryo-ET annotation is timely and valuable, and machine-learning approaches are likely to be important for speeding up the process, so I very much welcome this initiative. The authors have put a lot of thought and effort into the choice and acquisition of suitable sample data, including developing a new workflow with more-expensive computational tools to create a subset of reliable gold-standard or ground truth annotations. They have also put careful consideration into how to reduce the barriers of accessing the large volumes of data involved for participants. By getting participants to enter by submitting open-source software, they have ensured that any resulting methods will be readily accessible, and they have also side-stepped many of the complications (e.g. badly-formatted entries) arising when assessing other challenges, such as CASP. In addition, this mechanism allows them to standardize the measurement of run time. I expect that this challenge, and its subsequent iterations, will play a substantial role in stimulating new developments in structural biology at the cellular level.

There are a few issues that raise questions and minor concerns. The challenge described in this manuscript was opened to the community on 6 November 2024 and will close on 5 February 2025. This means that the paper is rather unlikely to be published before the entries have all been submitted and evaluated. So the timing is somewhat surprising: it is too late for referee comments to have any impact on the choices made in setting up the challenge, and it is a bit too early to gauge the success of the initiative in terms of numbers or quality of responses. I wouldn't want to argue that the manuscript should not be published now, given the amount of work it represents and the contributions that have been made to streamlining more conventional workflows, but I probably would have considered aiming for a special issue of a suitable journal that includes publication of results.

It is too late for this challenge, but there are some suggestions I would like the authors to consider for a followup. My feeling is that they have made unnecessary limiting assumptions about how machine-learning algorithms would work and be able to operate within the time limits they have set. For different aspects of the ground-truth annotations, they have used paired tomograms, tomograms before and after CTF correction, tomograms before and after denoising, and finer (5 Å) voxel sizes. However, for test data the participants in the challenge are only given tomograms that have had CTF correction and denoising applied, with 10 Å voxel sizes. This rules out any algorithms that might benefit from comparison of paired tomograms or from the weak signal that is present at higher resolution.

Finally, the authors might feel that the scripts they have developed and revised for the gold-standard annotations are relatively less important, but I think they should at least be mentioned in the abstract.

Randy Read

Reviewer #2:

Remarks to the Author:

Summary of key results: The authors present the methods they developed to create a machine learning tomogram particle-picking/annotation challenge. The results include (1) development of eight new software tools for particle picking, image/volume visualization, and annotation curation, (2) development of a novel method for creating a specimen containing a standardized set of particles within a cell lysate ("phantom sample") that provides suitable tomograms for a community-wide challenge activity, (3) creation of a curated, high-quality "ground truth" dataset of particle coordinates and assignments for 492 individual tomograms from the phantom sample specimen, (4) development of a statistical method to evaluate how well each challenge submission does in comparison to the ground truth dataset.

Originality and significance: The sample "Target" is novel in having both standardized particles and non-standardized cellular components. The result would be expected to be a highly attractive combo for testing and training machine learning algorithms for particle picking and classification, as both standardized and non-standardized components can be evaluated from the same tomograms. Important to note that the methods/results described are currently in use in an open Kaggle community challenge (running through Feb 2025). Over 400 teams have already submitted data / are listed on the leaderboard.

Validity of approach, quality of data, quality of presentation. The authors' proposed approach to solving the need for better machine learning algorithms for particle recognition/labelling of cellular tomograms is to create an attractive, open community competition. What is described in this manuscript is how the authors prepared samples/tomograms/data/comparison methods for the competition, and the process was clearly a major research project unto itself. The logic behind the approach to generating the challenge materials is strong and is well presented in the text and figures. The figures are well designed and intuitive.

Detailed comments/suggestions follow below.

Abstract line 19: perhaps imaging or exploring rather than understanding

Abstract line 27: spell out ML (especially since it can mean either Machine Learning or Maximum Likelihood).

lines 242, 273 CNN needs to be defined

line 246 Meaning Unclear: "First, the model was switched from a U-Net to Residual U-Net architecture for more stable training." e.g., what is a U-Net?

line 286: my guess is that the intended meaning is that false negatives ARE missed particles, but sentence reads as if they are two different categories.

lines 251, 288: Please define/describe what is meant by expert annotators/experts. (humans? how was expertise attained?)

line 330: incentivizing (motivating) doesn't seem like the right word.

line 344: Are x,y,z coordinates to be provided in pixels/voxels, Angstroms, or nanometers from origin position?

line 524: were specimen preparation pipetting steps performed manually by a human?

Table 1/line 495: suggest to provide primary citations for each EMDB map.

line 540: unclear what is meant by per-species error, and how this error was determined is also not described, but should be. Does value of 15% mean that the concentration was determined to be 15% of target concentration, or is the concentration 15% lower (or higher?) than target concentration?

General comment: It was unclear to me from main text whether participants are to submit only coordinates/assignments OR must they also submit their machine learning model algorithms (is this what is meant by "submitted model"?).

GFP nanobody: is this a nanobody with a GFP tag, or a nanobody that recognizes/binds to GFP tags (in which case perhaps refer to it as anti-GFP nanobody)?

line 371/372: public evaluation vs private evaluation—please explain rationale for these two evaluation steps.

line 830 Fig 1 legend title could be more descriptive — e.g. Cryo-electron tomography of challenge targets or challenge sample.

line 838 SNR is not defined

line 841 suggest to indicate typical angular range of tilt

line 853 example particle position...

Fig 1 panel C: Arrows are a little hard to see in the denoised image.

line 905 either in the Figure 4 legend or in the main text: briefly describe what is meant by leaderboard (I finally figured it out after visiting the kaggle competition page, but was confused what that meant before I did so).

Reviewer #3:

Remarks to the Author:

This manuscript presents a curated "ground truth" dataset for a contest that is currently open to the public through CZI. The work required is an impressive amount of software implementation/development, along with much hand-annotation, and the dataset will likely provide inspiration for algorithm development in the field, given that few "real" annotated datasets exist. That being said, it's not clear what utility the publication would play as a Resource in Nature Methods. In its current form, it reads a little more like a promise of things to come than it does as a true resource for the community. The really important things required for evaluating the quality of the ground truth are missing in my opinion. For instance, the in-house tools developed while generating the ground truth are not fully described, and it is explicitly stated that they will be in future publications. Unfortunately, this limits my understanding of the ground truth. Additionally, it is not clear from the results or the methods how each of the results from different softwares or hand annotations were merged and validated. Finally, (and of less concern) the "phantom" nature of the dataset will likely limit the effectiveness of tools developed around it for annotating real cellular tomograms. Because of these reasons, I cannot recommend this for publication in Nature Methods, as is.

I believe, for instance, including a full dissection and disclosure of the competition results, would go a long way toward making this a useful resource as it would set the state-of-the-art and provide a target for the community to aim for moving forward.

** For Nature Portfolio general information and news for authors, see <http://npg.nature.com/authors>.

Version 1:

Decision Letter:

31st Jan 2025

Dear Reza,

It was nice to chat with you yesterday. Thank you for your email and discussion asking us to reconsider our decision on your Resource, "Annotating CryoET Volumes: A Machine Learning Challenge". We have decided that we are willing to consider a revised version of your manuscript that is reformatted as a Brief Communication (roughly 1500 words/2 figures) and focuses on the characterization of the dataset.

- * include a point-by-point response to our referees and to any editorial suggestions
- * please underline/highlight any additions to the text or areas with other significant changes to facilitate review of the revised manuscript
- * address the points listed described below to conform to our open science requirements
- * ensure it complies with our general format requirements as set out in our guide to authors at www.nature.com/naturemethods
- * resubmit all the necessary files electronically by using the link below to access your home page

Link Redacted

We hope to receive your revised paper within 4 weeks. If you cannot send it within this time, please let us know. In this event, we will still be happy to reconsider your paper at a later date so long as nothing similar has been accepted for publication at Nature Methods or published elsewhere.

OPEN SCIENCE REQUIREMENTS

REPORTING SUMMARY AND EDITORIAL POLICY CHECKLISTS

When revising your manuscript, please submit reporting summary and editorial policy checklists.

DATA AVAILABILITY

CODE AVAILABILITY

Please include a "Code Availability" subsection in the Online Methods which details how your custom code is made available. Only in rare cases (where code is not central to the main conclusions of the paper) is the statement "available upon request" allowed (and reasons should be specified).

MATERIALS AVAILABILITY

ORCID

Sincerely yours,
Allison

Allison Doerr, Ph.D.
Chief Editor
Nature Methods

Version 2:

Decision Letter:

Our ref: NMETH-BC58164B

28th Mar 2025

Dear Reza,

Thank you for submitting your revised manuscript "A Real-World Phantom Dataset to Spur Innovation in CryoET Data Annotation" (NMETH-BC58164B). It has now been seen by the original referees and their comments are below. The reviewers find that the paper has improved in revision, and therefore we'll be happy in principle to publish it in Nature Methods, pending minor revisions to satisfy the referees' final requests and to comply with our editorial and formatting guidelines.

TRANSPARENT PEER REVIEW

ORCID

Sincerely yours,
Allison

Allison Doerr, Ph.D.
Chief Editor
Nature Methods

Reviewer #1 (Remarks to the Author):

My comments on the original manuscript focused largely on the timing of publishing a paper on their cryoET annotation challenge, which would appear after the challenge had ended but without discussing the results. The revisions have resulted in a shorter more focused paper on the development of the data resource underlying the challenge, and its continuing importance to the development of improved methods.

I think this is a reasonable response by the authors. I see that Nature Methods does publish papers about resources, either as special resource papers or as brief communications, and I believe that this resource will be useful enough to the community over the next few years to justify its publication.

Randy Read

Reviewer #3 (Remarks to the Author):

I believe that the authors have adequately addressed the concerns raised and I see no issues publishing this as a brief communication.

Version 3:

Decision Letter:

24th Jul 2025

Dear Reza,

I am pleased to inform you that your Brief Communication, "A Realistic Phantom Dataset for Benchmarking CryoET Data Annotation", has now been accepted for publication in Nature Methods. The received and accepted dates will be 4 November 2024 and 24 July 2025. This note is intended to let you know what to expect from us over the next month or so, and to let you know where to address any further questions.

Over the next few weeks, your paper will be copyedited to ensure that it conforms to Nature Methods style. Once your paper is typeset, you will receive an email with a link to choose the appropriate publishing options for your paper and our Author Services team will be in touch regarding any additional information that may be required.

Once proofs are generated, they will be sent to you electronically and you will be asked to send a corrected version within 48

hours. It is extremely important that you let us know now whether you will be difficult to contact over the next month. If this is the case, we ask that you send us the contact information (email, phone and fax) of someone who will be able to check the proofs and deal with any last-minute problems.

If, when you receive your proof, you cannot meet the deadline, please inform us at rjsproduction@springernature.com immediately.

If you are active on X or Bluesky, please e-mail me your and your coauthors' handles so that we may tag you when the paper is published.

To assist our authors in disseminating their research to the broader community, our SharedIt initiative provides you with a unique shareable link that will allow anyone (with or without a subscription) to read the published article. Recipients of the link with a subscription will also be able to download and print the PDF. As soon as your article is published, you will receive an automated email with your shareable link.

Please note that you and your coauthors may order reprints and single copies of the issue containing your article through Springer Nature Limited's reprint website, which is located at <http://www.nature.com/reprints/author-reprints.html>. If there are any questions about reprints please send an email to author-reprints@nature.com and someone will assist you.

Best regards,
Allison

Allison Doerr, Ph.D.
Chief Editor
Nature Methods

** Visit the Springer Nature Editorial and Publishing website at http://www.springernature.com/editorial-and-publishing-jobs?utm_source=ejP_NMeth_email&utm_medium=ejP_NMeth_email&utm_campaign=ejp_Nmeth for more information about our career opportunities. If you have any questions please click [here](mailto:editorial.publishing.jobs@springernature.com).

Reviewers' Comments:

Reviewer #1:

Remarks to the Author:

The authors present the organizational framework of a community challenge they have
prepared, addressing the need for faster and more reliable ways to annotate tomographic
reconstructions from cryo-EM (cryo-ET). Cryo-ET technology is improving rapidly and becoming
more widely available, and annotation is key to using it to understand the 3D organization of
cells and tissues.

The idea of a challenge for cryo-ET annotation is timely and valuable, and machine-learning
approaches are likely to be important for speeding up the process, so I very much welcome this
initiative. The authors have put a lot of thought and effort into the choice and acquisition of
suitable sample data, including developing a new workflow with more-expensive computational
tools to create a subset of reliable gold-standard or ground truth annotations. They have also
put careful consideration into how to reduce the barriers of accessing the large volumes of data
involved for participants. By getting participants to enter by submitting open-source software,
they have ensured that any resulting methods will be readily accessible, and they have also
side-stepped many of the complications (e.g. badly-formatted entries) arising when assessing
other challenges, such as CASP. In addition, this mechanism allows them to standardize the
measurement of run time. I expect that this challenge, and its subsequent iterations, will play a
substantial role in stimulating new developments in structural biology at the cellular level.

There are a few issues that raise questions and minor concerns. The challenge described in this
manuscript was opened to the community on 6 November 2024 and will close on 5 February
2025. This means that the paper is rather unlikely to be published before the entries have all
been submitted and evaluated. So the timing is somewhat surprising: it is too late for referee
comments to have any impact on the choices made in setting up the challenge, and it is a bit too
early to gauge the success of the initiative in terms of numbers or quality of responses. I
wouldn't want to argue that the manuscript should not be published now, given the amount of
work it represents and the contributions that have been made to streamlining more conventional
workflows, but I probably would have considered aiming for a special issue of a suitable journal
that includes publication of results.

**Response:** We thank the reviewer for raising this concern. Given the timing and to address this
concern, we have chosen to resubmit our manuscript as a Brief Communication that focuses
fully on the design and annotation of our phantom dataset and only briefly alludes to the
challenge. Initially our top priority for this manuscript was providing a good reference for the

participants of the challenge, who are mostly machine learning experts with little to zero domain
knowledge in cryoET. It seems that we were successful in regards to that aim as one can see
from the discussions of the paper on the Kaggle forum, and the fact that the paper has been
downloaded 2,244 times as of Feb 10th. We also acknowledge that as a standalone paper it
needs significant revision, in line with the reviewer's concern. We have now completely
reformatted the manuscript with a focus on describing the dataset, and we have mostly removed
the sections related to the challenge. We believe that this standalone paper will provide a
valuable resource for many groups over an extended period of time as we gradually improve
picking methods for cellular tomography. As a result, we believe that the details describing the
Kaggle challenge and its outcome belong in a separate paper.

It is too late for this challenge, but there are some suggestions I would like the authors to
consider for a followup. My feeling is that they have made unnecessary limiting assumptions
about how machine-learning algorithms would work and be able to operate within the time limits
they have set. For different aspects of the ground-truth annotations, they have used paired
tomograms, tomograms before and after CTF correction, tomograms before and after denoising,
and finer (5 Å) voxel sizes. However, for test data the participants in the challenge are only
given tomograms that have had CTF correction and denoising applied, with 10 Å voxel sizes.
This rules out any algorithms that might benefit from comparison of paired tomograms or from
the weak signal that is present at higher resolution.

**Response:** We appreciate the reviewer's concern. The reason that the test dataset in the
challenge contains only denoised tomograms at 10Å resolution is in fact a Kaggle limitation
arising from concerns about hosting multiple tomogram types for 492 tomograms. However, for
the training set that contained 7 tomograms we were able to provide multiple reconstruction
types such as: WBP, CTF-deconvolved, IsoNet-corrected, and denoised. Now that the Kaggle
challenge is complete, we are releasing all the data to the CryoET Data Portal, starting from
frames and including different reconstruction types along with all the alignment information. We
think this data will serve as a highly valuable resource for general particle annotation algorithms
well beyond the current Kaggle challenge.

Finally, the authors might feel that the scripts they have developed and revised for the gold-
standard annotations are relatively less important, but I think they should at least be mentioned
in the abstract.

**Response:** We thank the reviewer for this feedback. We have now revised the abstract to
include these workflows.

**Reviewer #2:**

**Remarks to the Author:**

Summary of key results: The authors present the methods they developed to create a machine
learning tomogram particle-picking/annotation challenge. The results include (1) development of
eight new software tools for particle picking, image/volume visualization, and annotation
curation, (2) development of a novel method for creating a specimen containing a standardized
set of particles within a cell lysate (“phantom sample”) that provides suitable tomograms for a
community-wide challenge activity, (3) creation of a curated, high-quality “ground truth” dataset
of particle coordinates and assignments for 492 individual tomograms from the phantom sample
specimen, (4) development of a statistical method to evaluate how well each challenge
submission does in comparison to the ground truth dataset.

Originality and significance: The sample “Target” is novel in having both standardized particles
and non-standardized cellular components. The result would be expected to be a highly
attractive combo for testing and training machine learning algorithms for particle picking and
classification, as both standardized and non-standardized components can be evaluated from
the same tomograms. Important to note that the methods/results described are currently in use
in an open Kaggle community challenge (running through Feb 2025). Over 400 teams have
already submitted data / are listed on the leaderboard.

Validity of approach, quality of data, quality of presentation. The authors’ proposed approach to
solving the need for better machine learning algorithms for particle recognition/labelling of
cellular tomograms is to create an attractive, open community competition. What is described
in this manuscript is how the authors prepared samples/tomograms/data/comparison methods
for the competition, and the process was clearly a major research project unto itself. The logic
behind the approach to generating the challenge materials is strong and is well presented in the
text and figures. The figures are well designed and intuitive.

**Detailed comments/suggestions follow below.**

**Abstract line 19: perhaps imaging or exploring rather than understanding**

**Done.**

**Abstract line 27: spell out ML (especially since it can mean either Machine Learning or**
**Maximum Likelihood).**

**Done.**

**lines 242, 273 CNN needs to be defined**

**Done. Moved to the Methods section.**

**line 246 Meaning Unclear: “First, the model was switched from a U-Net to Residual U-Net**
**architecture for more stable training.” e.g., what is a U-Net?**

**Done. Moved to the Methods section. Added a sentence and a reference.**

**line 286: my guess is that the intended meaning is that false negatives ARE missed particles,**
**but sentence reads as if they are two different categories.**

This section has been modified extensively to bring clarity to the process of ground truth
generation.
lines 251, 288: Please define/describe what is meant by expert annotators/experts. (humans?
how was expertise attained?)
We have added “human” to “expert annotators” and we now have a section describing this: “It is
important to note that through multiple iterations of this process, the human experts developed a
familiarity with the shape of the particles in the dataset, enabling them to manually curate the
final picks after working on this problem for several months.”
line 330: incentivizing (motivating) doesn’t seem like the right word.
This section no longer exists.
line 344: Are x,y,z coordinates to be provided in pixels/voxels, Angstroms, or nanometers from
origin position?
This section no longer exists.
line 524: were specimen preparation pipetting steps performed manually by a human?
Yes.
Table 1/line 495: suggest to provide primary citations for each EMDB map.
We now reference the primary citations in this Table.
line 540: unclear what is meant by per-species error, and how this error was determined is also
not described, but should be. Does value of 15% mean that the concentration was determined
to be 15% of target concentration, or is the concentration 15% lower (or higher?) than target
concentration?
Edit for clarity:
“The target concentration (after the 6-fold dilution due to mixing) for all of the purified species
(except for HSA) was 5 μ M. These are the deficiencies in hitting the 5 μ M target, calculated as
percentages ((5 μ M - measured concentration)/5 μ M): 1) THG: 15%, 2) Apoferritin: 33%, 3) Beta-
galactosidase: 27%, 4) Beta-amylase: 16%. HSA concentration was kept high because it served
as a background protein. VLP concentration could not be pushed further because of difficulties
with volume handling at high concentrations.”
General comment: It was unclear to me from main text whether participants are to submit only
coordinates/assignments OR must they also submit their machine learning model algorithms (is
this what is meant by “submitted model”?).
This section no longer exists since our manuscript no longer describes the Kaggle challenge in
depth.
GFP nanobody: is this a nanobody with a GFP tag, or a nanobody that recognizes/binds to GFP
tags (in which case perhaps refer to it as anti-GFP nanobody)?
This was the latter, and we have revised the description accordingly.
line 371/372: public evaluation vs private evaluation—please explain rationale for these two
evaluation steps.
This section no longer exists.
line 830 Fig 1 legend title could be more descriptive — e.g. Cryo-electron tomography of
challenge targets or challenge sample.
We have revised this figure in line with the shifted focus of our revised manuscript.
line 838 SNR is not defined

This is now defined in the Introduction.
line 841 suggest to indicate typical angular range of tilt
The possible angular range and range used to collect the phantom are now specified.
line 853 example particle position...
The relevant figure legend has been revised for clarity.
Fig 1 panel C: Arrows are a little hard to see in the denoised image.
This panel no longer exists in the revised figure.
line 905 either in the Figure 4 legend or in the main text: briefly describe what is meant by
leaderboard (I finally figured it out after visiting the kaggle competition page, but was confused
what that meant before I did so).
This section no longer exists.

Reviewer #3:

Remarks to the Author:

This manuscript presents a curated “ground truth” dataset for a contest that is currently open to
the public through CZI. The work required is an impressive amount of software
implementation/development, along with much hand-annotation, and the dataset will likely
provide inspiration for algorithm development in the field, given that few “real” annotated
datasets exist.

That being said, it’s not clear what utility the publication would play as a Resource in Nature
Methods. In its current form, it reads a little more like a promise of things to come than it does
as a true resource for the community.

**Response:** We appreciate the reviewer’s concern. We have addressed a similar concern from
review #1. Briefly, we wrote this manuscript to help the challenge participants get a solid footing.
However, we now understand that the manuscript needs revision to be standalone in the long-
term. We have revised the manuscript into a brief communications format with a focus on
describing the dataset. With the challenge done, the complete dataset is now available on the
CryoET Data Portal. So the concern about delivering on the promise should be resolved now.

The really important things required for evaluating the quality of the ground truth are missing in
my opinion. For instance, the in-house tools developed while generating the ground truth are not
fully described, and it is explicitly stated that they will be in future publications. Unfortunately,
this limits my understanding of the ground truth.

**Response:** We thank the reviewer for this feedback. This comment caught us off-guard given
that the original manuscript dedicated a full section, one main figure, and two supplemental
figures to describing how the ground truth was generated and its quality assessed. As our
manuscript will be resubmitted as a brief communications, we have had to abridge this section
in the main text. However, detailed descriptions of both the in-house tools and per-particle
strategies developed to generate the ground truth are provided in the Methods section.
Importantly, our manuscript now includes subtomogram averaging results for each species in

Fig. 1 so that readers can more easily assess the quality of the ground truth. We also provide
representative 2D classes in the supplemental figures since the final rounds of 2D classification
and manual inspection were the most crucial steps in generating clean sets of picks. As the
value of our annotated phantom as a resource for the community depends more on the quality
of the picks rather than how they were generated, our revisions prioritize the former.

**Additionally, it is not clear from the results or the methods how each of the results from different**
**softwares or hand annotations were merged and validated.**

**Response:** Merging was achieved by a combination of the copick and slabpick tools developed
as part of this work. Copick provided a common framework for storing and accessing particle
coordinates, including different versions of coordinates as they were updated throughout the
curation process. As a result, we adapted all of our methods to be able to convert coordinates to
copick format. Slabpick offered tools to merge sets of coordinates generated by distinct software
packages, with the distance threshold used for duplicate removal dependent on the particle
size. Initial validation was primarily done by visual inspection and 2D classification in
CryoSPARC, with slabpick used to generate the input per-particle 2D projections and extract the
selected particles for storage in copick format. Final validation also included subtomogram
averaging results for each of the six particles. We have further clarified these points in the
Methods and now include the 3D reconstructions in Fig. 1.

**Finally, (and of less concern) the “phantom” nature of the dataset will likely limit the**
**effectiveness of tools developed around it for annotating real cellular tomograms. Because of**
**these reasons, I cannot recommend this for publication in Nature Methods, as is.**

**Response:** We fully recognize that cellular tomograms present a distinct challenge because of
their molecular crowdedness, which the lysate and HSA included in our phantom could not fully
replicate. However, providing an effective testbed for annotation algorithms comes with two
significant constraints. First, the dataset has to be sufficiently large so that hundreds of
tomograms can be withheld as test data to rigorously benchmark algorithms and prevent
overfitting. Second, the ground truth annotations provided as training data must be highly
accurate and include diverse targets so that algorithms are encouraged to generalize across
molecular species. For an in situ sample, fulfilling the first constraint would have required years
of effort, while the second is out of reach of any current technology. Our phantom dataset
required considerable effort but enabled us to achieve both within months. Further, we
experienced firsthand while generating the ground truth that current algorithms perform poorly in
terms of speed, adaptability, and generalizability even for a less crowded sample like the
phantom. Therefore, we consider our phantom dataset a critical first step toward resolving these
issues.

**I believe, for instance, including a full dissection and disclosure of the competition results, would**
**go a long way toward making this a useful resource as it would set the state-of-the-art and**
**provide a target for the community to aim for moving forward.**

**Response:** We agree with the reviewer that the results of the challenge will be informative and
exciting, but we think that such an analysis requires an independent manuscript that can get into
all the details and focus on the challenge results. The current manuscript instead provides a full
description of the data that is now on the CryoET Data Portal, which will provide a gold standard
ground truth to benchmark future particle annotation algorithms.